# A NEURAL DIFFERENCE-OF-ENTROPIES ESTIMATOR FOR MUTUAL INFORMATION

## ABSTRACT

Estimating Mutual Information (MI), a key measure of dependence of random quantities without specific modelling assumptions, is a challenging problem in high dimensions. We propose a novel mutual information estimator based on parametrizing conditional densities using normalizing flows, a deep generative model that has gained popularity in recent years. This estimator leverages a block autoregressive structure to achieve improved bias-variance trade-offs on standard benchmark tasks.

## 1 INTRODUCTION

Mutual Information (MI), a measure of dependence of random variables $X$ and $Y$, plays an important role in information theory (Cover & Thomas, 2006), statistics and machine learning (Tishby et al., 2000; Peng et al., 2005; Vergara & Estévez, 2014; Chen et al., 2018), and biology and medical sciences (Zhang et al., 2012; Sengupta et al., 2022). For random variables $X$ and $Y$ with joint density $p$, the mutual information is defined as

$$I(X;Y) = D_{\mathrm{KL}}(p \parallel p_X \otimes p_Y) = \mathbb{E}_{(X,Y) \sim p} \left[ \log \frac{p(X,Y)}{p_X(X)p_Y(Y)} \right],$$

where $p_X$ and $p_Y$ are the marginal densities of $X$ and $Y$, $p_X \otimes p_Y$ is the density of the product distribution, and $D_{\mathrm{KL}}(\cdot \parallel \cdot)$ is the Kullback-Leibler (KL) divergence. We consider the problem of estimating mutual information from a finite set of samples $\{(x_1, y_1), \ldots, (x_N, y_N)\}$.

Formally, an MI estimator $\hat{I}_N$ depends on independent and identically distributed (i.i.d.) random sample pairs $(X_1, Y_1), \ldots, (X_N, Y_N)$, and should ideally be unbiased, consistent and efficient. In addition, such an estimator should be effectively computable from large and high-dimensional data. We propose an unbiased and consistent mutual information estimator based on the difference-of-entropies (DoE) estimator suggested in McAllester & Stratos (2018). This characterization expresses the mutual information as the difference between the entropy of $X$ and the conditional entropy of $X$ given $Y$,

$$I(X;Y) = H(X) - H(X \mid Y).$$

Each of the terms in this expression can be characterized as the infimum of a variational optimization problem. Our implementation of this estimator is based on carefully chosen normalizing flows that simultaneously approximate the minimizing densities of each of the optimization problems.

### 1.1 OVERVIEW OF PREVIOUS WORK

Traditional MI estimators are non-parametric estimators that depend on density estimation and Monte Carlo integration or on the computation of $k$ nearest neighbours (kNN). Examples include the widely used KSG estimator by Kraskov et al. (2004), the non-parametric kNN estimator (kpN) by Lombardi & Pant (2015), and improvements of the KSG estimator and a geometric kNN estimator by Gao et al. (2017). These non-parametric methods are fast and accurate for low-dimensional and small-sized problems and are easy to implement. However, they suffer from the curse of dimensionality and do not scale well in machine learning problems since the data sets can be relatively large and high-dimensional (Paninski, 2003).

More recent parametric methods take advantage of deep learning architectures to approximate variational bounds on MI. These have been categorized into discriminative and generative approaches

by Song & Ermon (2019). Some state-of-art discriminative approaches include InfoNCE (van den Oord et al., 2018), MINE (Belghazi et al., 2018), SMILE (Song & Ermon, 2019), CCMI (Mukherjee et al., 2019) and DEMI (Liao et al., 2020). van den Oord et al. (2018) proposed a contrastive predictive coding (CPC) method which relies on maximizing a lower bound on mutual information. The lower bound involves function approximators implemented with neural networks and is constrained by the batch size $N$, leading to a method that is more biased but with less variance. MINE, on the other hand, is based on the Donsker-Varadhan (DV) lower bound for the KL divergence. Fundamental limitations on approaches based on variational bounds were studied by Song & Ermon (2019) and McAllester & Stratos (2018), the latter being the motivation for our approach.

Instead of constructing the Mutual Information estimators based on variational lower bounds, Liao et al. (2020) proposed a classifier-based estimator called DEMI, where a parametrized classifier is trained to distinguish between the joint density $p(x, y)$ and the product $p(x)p(y)$. Mukherjee et al. (2019) proposed another classifier-based (conditional) MI estimator that is asymptotically equivalent to DEMI. However, it still relies on variational lower bounds and is prone to higher error than DEMI for finite samples, as summarized by Liao et al. (2020).

Compared with discriminative approaches, generative approaches are less well explored in MI estimation problems. A naïve approach using generative models for estimating MI is to learn the entropies $H(X), H(Y)$ and $H(X, Y)$ with three individual generative models, such as VAE (Kingma & Welling, 2013) or Normalizing Flows (Kobyzev et al., 2020), from samples. A method for estimating entropy using normalizing flows was introduced by Ao & Li (2022). Estimators based on the individual entropies will be highly biased and computationally expensive since the entropies are trained separately, while it is revealed that considering the enhancement of correlation between entropies in constructing MI estimators can improve the bias (Gao et al., 2017). Duong & Nguyen (2023) propose the Diffeomorphic Information Neural Estimator (DINE), that takes advantage of the invariance of conditional mutual information under diffeomorphisms. An alternative approach to estimating MI using normalizing flows was recently proposed by Butakov et al. (2024). This approach takes advantage of the invariance of the pointwise mutual information under diffeomorphisms. In addition, the methods from Butakov et al. (2024) allow for the estimation of mutual information using direct, closed-form expressions. Finally, we would like to point the recently introduced MINDE estimator Franzese et al. (2024), which is based on diffusion models and represents a complementary approach.

From a practical point of view, the performance of MI estimators is often measured using standard data sets based on Gaussian distributions for which the ground truth is known. Recently, Czyż et al. (2023) proposed a collection of benchmarks to evaluate the performance of different MI estimators in more challenging environments.

### 1.2 NOTATION AND CONVENTIONS

The entropy of a random variable with density $p$ is defined as $H(X) = -\mathbb{E}[\log p(X)]$, and we sometimes write $H(p)$ to highlight the dependence on the density. Throughout this paper, we work with absolutely continuous random variables and distributions.

## 2 MUTUAL INFORMATION AND NORMALIZING FLOWS

We begin by introducing the characterization of mutual information in terms of entropy that forms the basis of our approach. We then introduce normalizing flows and discuss an implementation of our mutual information estimator.

### 2.1 MUTUAL INFORMATION AND ENTROPY

Given a pair of random variables $(X, Y)$, the conditional entropy of $X$ with respect to $Y$ is defined as $H(X|Y) = H(X, Y) - H(Y)$, where $H(X, Y)$ is the joint entropy of $(X, Y)$ (not to be confused with the cross-entropy, introduced below). The mutual information can be expressed in terms of

entropies via the 3H principle:

$$I(X;Y) = H(X) + H(Y) - H(X,Y)$$
$$\stackrel{(*)}{=} H(X) - H(X|Y).$$
(1)

The characterization (*) is the basis of the difference-of-entropies (DoE) estimator introduced by McAllester & Stratos (2018).

The entropy of a random variable can be characterized as the solution of a variational optimization problem involving the cross-entropy. The cross-entropy between random variables $X$ and $Y$ with densities $p$ and $q$, respectively, is defined as

$$Q(p,q) := -\mathbb{E}_p[\log q(X)].$$

One easily checks that the cross-entropy, entropy and KL divergence are related via

$$Q(p,q) = H(X) + D_{\mathrm{KL}}(p \parallel q).$$
(2)

The KL divergence is non-negative and satisfies $D_{\mathrm{KL}}(p \parallel q) = 0$ if and only if $p = q$ almost everywhere. A well-known consequence of this fact is the following characterization of the entropy of a random variable with density $p$:

$$H(X) = \inf_q Q(p,q),$$

where the infimum is taken over all probability densities $q$.

The conditional entropy $H(X|Y)$ is itself the cross-entropy of the conditional density $p_{X|Y} = p/p_Y$ with respect to the joint density $p$,

$$H(X|Y) = -\mathbb{E}_p\left[\log \frac{p(X,Y)}{p_Y(Y)}\right] = Q(p, p_{X|Y}).$$

Note that a conditional probability density is not a joint density, as it does not integrate to 1, but the definition of cross-entropy and KL-divergence still makes sense. The proof of the following result is simple and is included for reference.

**Lemma 2.1.** *Let $(X,Y)$ be a pair of random variable with joint density $p$. Then*

$$H(X|Y) = \inf_q Q(p,q),$$

*where the infimum is over all conditional densities, i.e., non-negative functions $q(x|y)$ such that $\int_x q(x|y)\,\mathrm{d}x = 1$ for all $y$.*

*Proof.* Let $q(x|y)$ be a conditional density. Then

$$
\begin{aligned}
Q(p,q) &= -\mathbb{E}_p[\log q(X|Y)] \\
&= H(X|Y) + \mathbb{E}_p[\log p(X,Y)] - \mathbb{E}_p[\log(q(X|Y)p_Y(Y))] \\
&= H(X|Y) - H(X,Y) + Q(p,\tilde{q}) \\
&= H(X|Y) + D_{\mathrm{KL}}(p \parallel \tilde{q}),
\end{aligned}
$$

where $\tilde{q}(x,y) = q(x|y) \cdot p_Y(y)$ is a probability density. By equation 2, $Q(p,\tilde{q}) \geq H(X,Y)$, with equality if and only if $\tilde{q} = p$, i.e., $q = p_{X|Y}$. □

As a consequence of the proof (or by direct inspection) we get the following observation.

**Corollary 2.2.** *Let $(X,Y)$ be a pair of random variables with density $p$ and let $q(x,y)$ be a probability density. Then*

$$Q(p, q/p_Y) = D_{\mathrm{KL}}(p \parallel q) + H(X|Y)$$

Together with the 3H principle, equation 1, we get

$$I(X;Y) = \inf_{q_X} Q(p_X, q_X) - \inf_{q_{X|Y}} Q(p, q_{X|Y}).$$
(3)

Given data $\{(x_i, y_i)\}_{i=1}^N$, the resulting difference-of-entropies (DoE) estimator, as suggested by McAllester & Stratos (2018), consists of minimizing the objectives

$$\hat{Q}(p_X, q_X) = -\frac{1}{N}\sum_{i=1}^N \log q_X(x_i), \quad \hat{Q}(p, q_{X|Y}) = -\frac{1}{N}\sum_{i=1}^N \log q_{X|Y}(x_i|y_i) \tag{4}$$

with respect to $q_X$ and $q_{X|Y}$. In our implementation of the DoE estimator, we parametrize these densities jointly, rather than separately, using block autoregressive normalizing flows.

## 2.2 NORMALIZING FLOWS

A popular way of estimating densities is via normalizing flows, where the density to be estimated is seen as the density of a push-forward distribution of a simple base distribution, and the transformation is implemented using invertible neural networks. Let $g\colon \mathbb{R}^n \to \mathbb{R}^n$ be a measurable function, and let $\mu$ be a probability measure. The push-forward measure $g_*\mu$ is defined as

$$g_*\mu(A) = \mu(g^{-1}(A))$$

for all measurable $A$. The density of a random variable $X$ that has the push-forward distribution of an absolutely continuous random variable $Z$ with density $p_Z$ with respect to a diffeomorphism $g$ is also absolutely continuous, with a density function $p_X$ given by

$$p_X(x) = p_Z(g^{-1}(x)) \cdot \left|\det \mathrm{d}g(g^{-1}(x))\right|^{-1},$$

where $\mathrm{d}g(z)$ denotes the differential of $g$ at $z$ (in coordinates, given by the Jacobian matrix).

It is known that any continuous distribution with density $p_X$ satisfying some mild conditions can be generated from the uniform distribution on a cube $[0, 1]^n$ (and hence, by invertibility, from any other distribution satisfying the same conditions) if the transformation $f$ can have arbitrary complexity (Bogachev et al., 2007). However, as is common with universal approximation results, this result does not translate into a practical recipe. A more practical approach is to use a composition of simple functions implemented by neural networks, which have sufficient expressive power. An obvious but important property of diffeomorphisms is that they are composable. Specifically, let $g_1, g_2, \ldots, g_K$ be a set of $K$ diffeomorphisms and denote by $g = g_K \circ g_{K-1} \circ \cdots \circ g_1$ the composition of these functions. The determinant of the Jacobian is then given by

$$\det \mathrm{d}g(z) = \prod_{i=1}^K \det \mathrm{d}g_i(z_i),$$

where $z_i = g_{i-1} \circ \cdots \circ g_1(z)$ for $i \geq 2$ and $z_1 = z$ and $z_{K+1} = x = g(z)$. Similarly, for the inverse of $f$, we have

$$g^{-1} = g_1^{-1} \circ \cdots \circ g_K^{-1},$$

and the determinant of the Jacobian is computed accordingly. Thus, we can construct more complicated functions with a set of simpler, bijective functions. The two crucial assumptions in the theory of normalizing flows are thus invertibility ($g^{-1}$ should exist) and simplicity (each of the $g_i$ should be simple in some sense). The inverse direction, $f = g^{-1}$, is called the normalizing direction: it transforms a complicated distribution into a Gaussian, or normal distribution. For completeness and reference, we reiterate the transformation rule in terms of the normalizing map:

$$\log p_X(x) = \log p_Z(f(x)) + \log|\det \mathrm{d}f(x)|. \tag{5}$$

Normalizing flows are fitted by minimizing the KL divergence between a model $p_X(x; \Theta)$ and an unknown target distribution $p_X^*(x)$ from which we only see samples. Here, the model parameters are denoted as $\Theta = \{\phi, \psi\}$, where $\phi$ are the parameters of the normalizing function $f_\phi$, and $\psi$ are the parameters of the base density $p_Z(z; \psi)$. Because the KL divergence is asymmetric, the order in which the probabilities are listed is important, which leads to two different cost functions, the forward and the reverse KL divergence. In our work, we only focus on the forward KL divergence $D_{\mathrm{KL}}\left(p_X^* \parallel p_X(\cdot; \Theta)\right)$ since it applies in situations when we have no way to evaluate the target density $p_X^*(x)$, but we have (or can generate) samples from the target distribution.

In light of equation 2, minimizing the forward KL divergence is equivalent to minimizing the cross-entropy

$$
\begin{aligned}
\mathcal{L}(\Theta) &:= Q(p_X^*, p_X(\cdot; \Theta)) = -\mathbb{E}_{p_X^*}[\log p_X(X; \Theta)] \\
&= -\mathbb{E}_{p_X^*}[\log p_Z(f_\phi(X); \psi) + \log|\det \mathrm{d}f_\phi(X)|].
\end{aligned}
\tag{6}
$$

Given a set of samples $\{x_j\}_{j=1}^N$ from $p_X^*(x)$, $\mathcal{L}(\Theta)$ can be estimated by replacing the expectation with the empirical mean, which leads to the cost function

$$
\hat{\mathcal{L}}(\Theta) := -\frac{1}{N} \sum_{j=1}^N (\log p_Z(f_\phi(x_j); \psi) + \log|\det \mathrm{d}f_\phi(x_j)|).
\tag{7}
$$

Equation 7 is a Monte Carlo estimate of the cross entropy between the target distribution and the model distribution. The cost function $\mathcal{L}(\Theta)$ is minimized when $p_X^* = p_X(\cdot; \Theta)$, and the optimal value is the entropy of $X$. If the model is expressive enough to characterize the target distribution, then minimizing 7 over the parameters yields an entropy estimator.

### 2.2.1 BLOCK AUTOREGRESSIVE FLOWS

Autoregressive flows (Kingma et al., 2016) are normalizing flows with the convenient property that their Jacobian is triangular. Block neural autoregressive flows (B-NAF), introduced by De Cao et al. (2019), are flows that are autoregressive and monotone, but that are implemented using a single neural network architecture, rather than relying on conditioner networks. More specifically, a block autoregressive flow is given as a sequence of transformations

$$
f \colon \mathbb{R}^d \to \mathbb{R}^{da_1} \to \cdots \to \mathbb{R}^{da_\ell} \to \mathbb{R}^d,
$$

where each $f^k \colon \mathbb{R}^{da_k} \to \mathbb{R}^{da_{k+1}}$ is given by $f^k(x) = \sigma(W^k x + b^k)$ with $\sigma$ a strictly increasing activation function, and $W^k$ is a block matrix of the form

$$
W^k = \begin{bmatrix}
g(B_{11}^{(k)}) & 0 & \cdots & 0 \\
B_{21}^{(k)} & g(B_{22}^{(k)}) & \cdots & 0 \\
\vdots & \vdots & \ddots & \vdots \\
B_{d1}^{(k)} & B_{d2}^{(k)} & \cdots & g(B_{dd}^{(k)})
\end{bmatrix},
$$

where each $B_{ij}^{(k)} \in R^{a_{k+1} \times a_k}$ and $g(x) = \exp(x)$ applied componentwise, to ensure that the entries are positive. We set $a_0 = a_{\ell+1} = 1$. It is not hard to see that the $i$-component of $f(x)$ only depends on $x_1, \ldots, x_i$. Since the product of block diagonal matrices with blocks of size $a \times b$ and $b \times c$, respectively, is block diagonal with size $a \times c$, the composition $f$ has lower triangular Jacobian with positive diagonal entries, and hence is invertible. The determinant of the triangular Jacobian matrix is the product of the diagonal entries $\partial f_i / \partial x_i$, each of which can be computed as product

$$
\frac{\partial f_i}{\partial x_i} = \prod_{k=0}^{\ell} g(B_{ii}^{(k)}).
$$

In practice, implementations of B-NAF use masked networks and gated residual connections to improve stability, but this does not alter the analysis. Just as with neural autoregressive flows, it can be shown that B-NAF are universal density estimators.

## 3 JOINT ESTIMATION OF MUTUAL INFORMATION

Our goal is to minimize the functions in equation 3, where the density $q_X(x)$ and the conditional density $q_{X|Y}(x|y)$ are parametrized using normalizing flows. We implement the difference of entropies (DoE) estimator by constructing a specific neural network structure that can estimate the two entropies in equation 4 in the same framework by "deactivating" the certain sub-network. Technically, this is implemented by using a mask to set the contributions coming from one part of the network to another to zero.

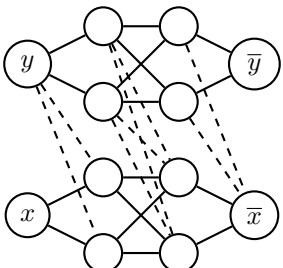

Figure 1: A Block Autoregressive Flow $f(y, x)$. Solid lines represent positive weights.

To motivate the architecture, consider the network in Figure 1, implementing a flow $f\colon \mathbb{R}^2 \to \mathbb{R}^2$ given as a composition $f = f^2 \circ f^1 \circ f^0$ with $f^0\colon \mathbb{R}^2 \to \mathbb{R}^4$, $f^1\colon \mathbb{R}^4 \to \mathbb{R}^4$ and $f^2\colon \mathbb{R}^4 \to \mathbb{R}^2$. Hence, $a_1 = a_2 = 2$ and the corresponding neural network has the form shown in Figure 1.

Recall that block autoregressive flows have the property that $f_i$ depends only on the first $i$ variables. In particular, we can express the function $f$ as

$$f(y, x) = (f_1(y), f_2(y, x)).$$

The Jacobian determinant is the product of the partial derivative $\partial f_1 / \partial y$ and $\partial f_2 / \partial x$ (see Section 2.2.1). Suppose $p(x, y)$ is a standard Gaussian density, so that $p(x, y) = p_X(x)p_Y(y)$, and that we have data $(x_i, y_i)$ from an unknown distribution $q$. The cost function equation 7 for learning a normalizing flow takes the form

$$-\frac{1}{N} \sum_{i=1}^{N} \left( \log p_X(f_2(y_i, x_i)) + \log \frac{\partial f_2}{\partial x}(y_i, x_i) \right) + \left( \log p_Y(f_1(y_i)) + \log \frac{\partial f_1}{\partial y}(y_i) \right). \quad (8)$$

The components $f_1$ and $f_2$ depend on a distinct set of weights in the neural network. Optimizing only the part of equation 8 involving $f_1$ on data $\{y_i\}$ gives an estimate for the entropy of $Y$, while optimizing the part with $f_2$ on data $\{(x_i, y_i)\}$ gives rise to an estimate of the cross-entropy $H(X \mid Y)$. Moreover, if we deactivate the weights in off-diagonal blocks (the dashed lines), then optimizing this part on data $\{x_i\}$ gives an estimate of $H(X)$. Note that training for $H(X \mid Y)$ and setting the off-diagonal weights to zero does not automatically give an estimators for $H(X)$. It is, however, conceivable that one can begin with a network that approximates $H(X)$ and then optimize the off-diagonal weights to obtain an approximation of $H(X|Y)$. A justification of this approach is provided in the appendix.

In general, we consider a flow $f\colon \mathbb{R}^{2n} \to \mathbb{R}^{2n}$ with a block autoregressive structure, given by $f(y, x) = (f_1(y), f_2(y, x))$ with $x \in \mathbb{R}^n$, $y \in \mathbb{R}^n$. The function $f_2$ is a composition of layers of the form

$$\sigma(W_{21}^{(\ell)} y^{(\ell-1)} + W_{22}^{(\ell)} x^{(\ell-1)} + b^{(\ell)}),$$

where $(y^{(\ell-1)}, x^{(\ell-1)})$ is the output of the previous layer of the flow $f$. Consider the cost function

$$\mathcal{L}_1 = -\frac{1}{N} \sum_{i=1}^{N} \left( \log p(f_2(y_i, x_i)) + \log \det |\mathrm{d}_x f_2(y_i, x_i)| \right),$$

where now we simply write $p$ for the density of a Gaussian. Optimizing this function gives an estimate of the cross-entropy $H(X \mid Y)$. If, on the other hand, we set the off-diagonal weights to zero and optimize the resulting function $\mathcal{L}_2$, we get an estimator for the entropy $H(X)$. This motivates Algorithm 1, which optimizes for $H(X \mid Y)$ and $H(X)$ simultaneously.

Algorithm 1 can be generalized to any normalizing flows with inner autoregressive structure between $X$ and $Y$. Compared with general autoregressive flows which usually model the autoregressive functions as conditioner neural networks, BNAF has not only the superior expressive power, but also the easy computation of Jacobian matrix and the straightforward deactivation operation given by the block-wise matrix form of autoregressive functions. The theoretical justification based on universal approximation results for Block Autoregressive Flows is provided in an appendix.

---

**Algorithm 1** Normalizing Flows MI Estimation

---

**Input:** data $(x_i, y_i)$
Initialize model parameters $\phi$.
**repeat**
  Draw minibatch $S$ of $M$ samples $\{(x_i, y_i)\}$
  Evaluate:

$$\mathcal{L}_1 = -\frac{1}{M} \sum_{(x,y) \in S} \left( \log p(f_2(y, x; \phi)) + \log|\det \mathrm{d}_x f_2(y, x; \phi)| \right)$$

  Update the parameters by gradients: $\phi = \phi + \nabla \mathcal{L}_1$
  Deactivate the off-diagonal weights, call new parameters $\phi'$
  Evaluate:

$$\mathcal{L}_2 = -\frac{1}{M} \sum_{(x,y) \in S} \left( \log p(f_2(y, x; \phi')) + \log|\det \mathrm{d}_x f_2(y, x; \phi')| \right)$$

  Update the parameters by gradients: $\phi' = \phi' + \nabla \mathcal{L}_2$
**until** Convergence
**Output:** $\hat{I}(X, Y) = \mathcal{L}_2 - \mathcal{L}_1$

---

## 4 NUMERICAL EXPERIMENTS

We implemented several experimental settings from prior work (Belghazi et al., 2018; Song & Ermon, 2019; Poole et al., 2019; Hjelm et al., 2018; Czyż et al., 2023) to evaluate the performance of the proposed estimator. We first focus on the accuracy of the resulting estimates on synthetic Gaussian examples, where the true value of MI can be calculated analytically. In the Appendix C, we report on additional experiments on extremely small-sized dataset and non-Gaussian distributions. The final experiments will be the long-run training behavior on the proposed estimator. All experiments were conduced on a computing cluster using Nvidia Quadro RTX 6000 GPUs.

### 4.1 MI ESTIMATION ON CORRELATED MULTIVARIATE GAUSSIANS

In this experiment, we sampled from two correlated Gaussian random variables $X$ and $Y$, for which the MI can be exactly obtained from their known correlation. The different MI estimators were trained on datasets with varying dimensionality of $X, Y$ (20-d, 50-d and 100-d), sample size (32K, 64K and 128K) and true MI to characterize the relative behaviors of every MI estimator. Additionally, we conduct an experiment by applying an element-wise cubic transformation on $y_i \to y_i^3$. This generates the non-linear dependencies in data without changing the ground truth of MI. The performance of trained estimators are evaluated on a different testing set of 10240 samples. Czyż et al. (Czyż et al., 2023) mentioned that Gaussians with sparse interactions between $X$ and $Y$ could be a challenging benchmark for MI estimations. We then sample from Gaussians with $\mathrm{Cor}(X1, Y1), \mathrm{Cor}(X2, Y2) > 0$ and there is no correlation between any other (distinct) variables. We named it Sparse Gaussian as the covariance matrix $Cov(X, Y)$ is sparse matricx in this case. We assessed our methods along with the following baselines: 1. **DEMI** (Liao et al., 2020), with the parameter $\alpha = 0.5$. 2. **SMILE** (Song & Ermon, 2019), with three clipping parameters $\tau \in \{5.0, \infty\}$. For $\tau = \infty$, it is equivalent to the MINE Belghazi et al. (2018); 3. **InfoNCE** (van den Oord et al., 2018), which is the method in contrastive predictive coding (CPC); 4. **NWJ** (Nguyen et al., 2010), which is the method that based on estimating the likelihood ratios by convex risk minimization; 5. **DoE** (McAllester & Stratos, 2018), the DoE method, where the distributions is parameterized by isotropic Gaussian (correct) or logistic (misspecified), with three parameters $\tau = 1.0$ that clips the gradient norm in training; 6. **BNAF**, approximating the entropies respectively in the MI using two separate Block Neural Autoregressive Flows; 7. **NDoE, BNAF**, the proposed method with BNAF structure; 8. **NDoE, Real NVP**, the proposed method with Real NVP structure. We noticed that the comparison between our method, as a generative model, and other discriminative methods can be difficult since the neural network structure and the model parametrizations are different. To make the comparison as fair as possible, we used the same neural network architecture for all discriminative

methods, which is a multi-layer perceptron with an initial concatenation layer, two fully connected layers with 512 hidden units for each layer and ReLU activations, and a linear layer with a single output. In terms of our proposed method, we constructed the flow with 2 BNAF transformation layers and tanh activations, and a linear BNAF layer to reset the dimensionality. The BNAF layers use $20 \times 20$-d, $10 \times 50$-d, $6 \times 100$-d hidden dimensions for 20-d, 50-d and 100-d data respectively, which is roughly the same as the 512 hidden units in discriminative methods. For Real NVP layers, we let each of the scale and translation functions to be two layers multi-layer perceptron with 128 hidden units for each layer and ReLU activations. Each MI estimator was trained for 50 epochs with a mini-batch of 128. Due to the vanishing and exploding gradient issues in Real NVP, we applied the Adamax optimizer with a fine-tuned learning rate. For all other optimizations, the Adam optimizer with a learning rate of 0.0005 was used. All results were computed over 10 runs on the testing sets generated with different random seeds to ensure robustness and generalizability.

**Results.** The results for a sample size of 128K are shown in Figure 2, while the results for sample sizes of 64K and 32K are presented in Figure 3 and Figure 4, respectively. The Sparse Gaussian results for the 20-dimensional case are plotted in Figure 5. Overall, all the discriminative methods tend to underestimate MI. This issue does not occur in our proposed flow-based models for Gaussian variables, likely due to the fact that the base distribution is itself Gaussian. For the cubic Gaussian case, the underestimation is much milder compared to other methods, though the underestimating bias increases as the true MI becomes larger.

Among all the methods, our proposed model achieved better performance across different dimensionalities and sample sizes. While **DoE** methods performed well for Gaussian variables, they exhibited a large bias when applied to cubic Gaussians. When comparing **NDoE, BNAF** with **BNAF**, we observed that for smaller sample sizes (or insufficient training steps), **BNAF** exhibits a slight bias across all true MI values. Additionally, for cubic cases, **BNAF** shows a larger bias when MI is close to zero, an issue not observed with **NDoE, BNAF**. The work by Song & Ermon (2019) attributed this as a shortcoming of generative models, but we believe our method mitigates this issue, as the bias in entropy estimation vanishes by approximating entropies using the same neural network.

In the cubic cases, **NDoE, Real NVP** exhibits a larger bias, though it still outperforms discriminative methods, particularly when the sample size is sufficiently large. In the 20-dimensional Gaussian case, **SMILE** occasionally overestimated MI, which we will further analyze in the long-run training experiments. For the Sparse Gaussian case, both **NDoE, BNAF** showed small biases when the true MI is small, and **BNAF** outperformed **NDoE, BNAF** for larger MI. Both methods consistently outperformed other approaches across different sample sizes. However, **NDoE, Real NVP** failed to achieve realistic results in the Sparse Gaussian case.

## 5 CONCLUSIONS

In this research, we proposed a new MI estimator which is based on the block autoregressive flow structure and the difference-of-entropies (DoE) estimator. Theoretically, our method converges to true mutual information as the number of samples increases and with large-enough neural network capacity. The accuracy of our estimator then depends on the ability of the block autoregressive flows in predicting the true posterior probability of items in the test set. A theoretical analysis is provided in the appendix. We discussed the connections and differences between our approach and other approaches including the lower bound approaches of MINE and SMILE and InfoNCE(CPC), and the classifier based approaches of CCMI and DEMI. We also demonstrate empirical advantages of our approach over the state of the art methods for estimating MI in synthetic data. Given its simplicity and promising performance, we believe that our method is a good candidate for use in research that optimizes MI. In future work, we aim to expand our experiments to additional data, including the image-like data of (Butakov et al., 2024) and the recently published benchmarks in (Lee & Rhee, 2024).

Despite its promising results, the proposed method has limitations in its current form. As a method that depends on the particular neural network architecture used to implement the flows, care needs to be taken to ensure the stability of the proposed estimator and its performance on smaller data sets. As seen in the experiments, the method performs particularly well in cases where the random quantities are based on Gaussian distribution. In future work we aim to explore the possibilities of using different

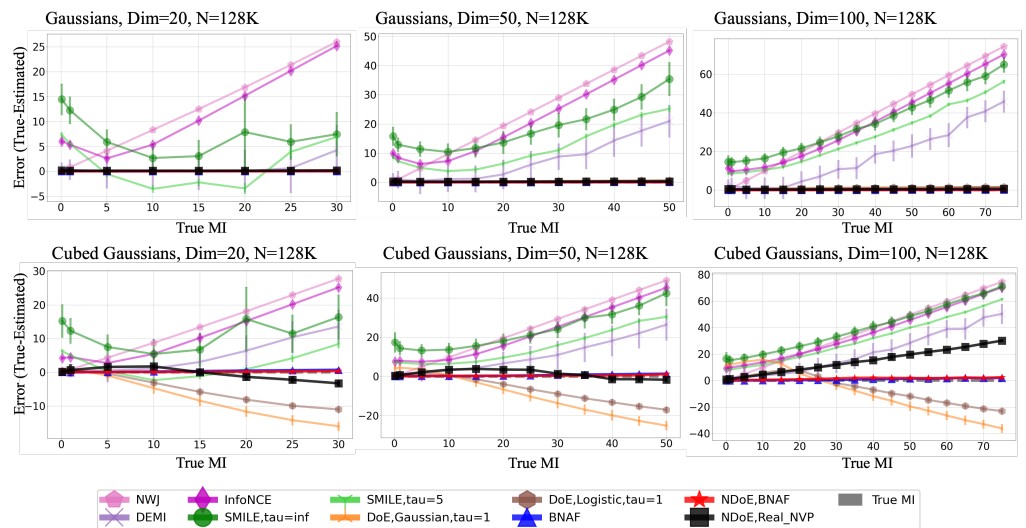

Figure 2: MI estimation between multivariate Gaussian variables (Top) and between multivariate Gaussian variables with a cubic transformation (Bottom). The size of training data are 128K. The estimation error $(I(x,y) - \hat{I}(x,y))$ are reported. Closer to zero is better.

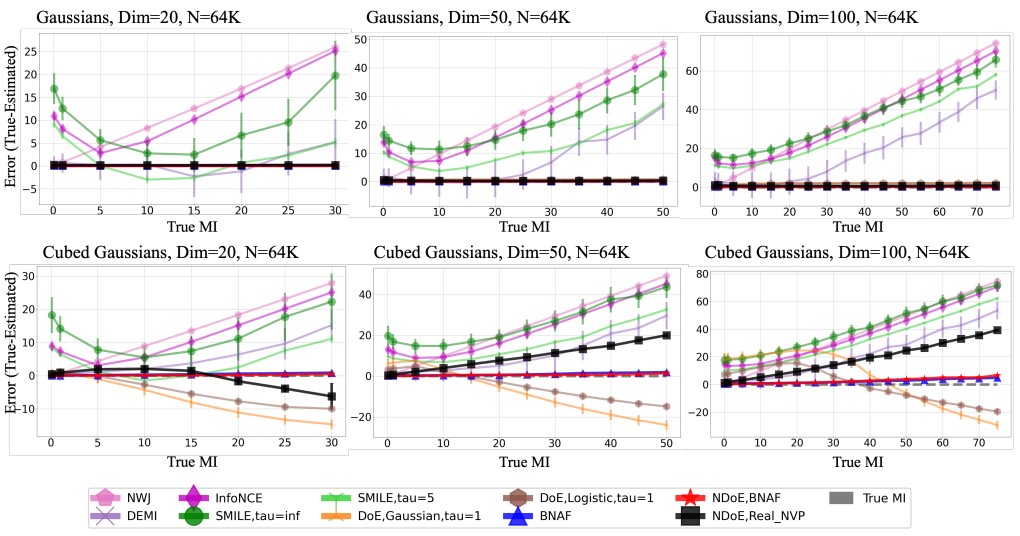

Figure 3: MI estimation between multivariate Gaussian variables (Top) and between multivariate Gaussian variables with a cubic transformation (Bottom). The size of training data are 64K. The estimation error $(I(x,y) - \hat{I}(x,y))$ are reported. Closer to zero is better.

classes of base distributions. This includes the possibility of dealing with discrete distributions, a situation that is handled well by critic-based methods Belghazi et al. (2018); van den Oord et al. (2018). Another direction involves evaluating our method in view of downstream applications that require the computation of mutual information and comparing its performance in these settings with other generative approaches that were recently introduced (Franzese et al., 2024; Duong & Nguyen, 2023; Butakov et al., 2024). In particular, in light of recent work Kong et al. (2023); Franzese et al. (2024), it would be interesting to explore multimodal examples, like the MI between image data and text embeddings.

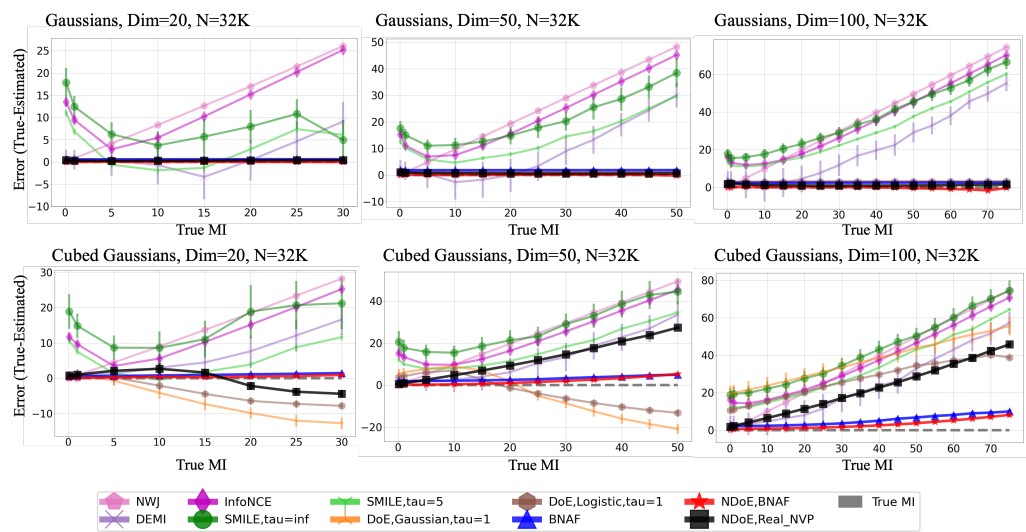

Figure 4: MI estimation between multivariate Gaussian variables (Top) and between multivariate Gaussian variables with a cubic transformation (Bottom). The size of training data are 32K. The estimation error $(I(x, y) - \hat{I}(x, y))$ are reported. Closer to zero is better.

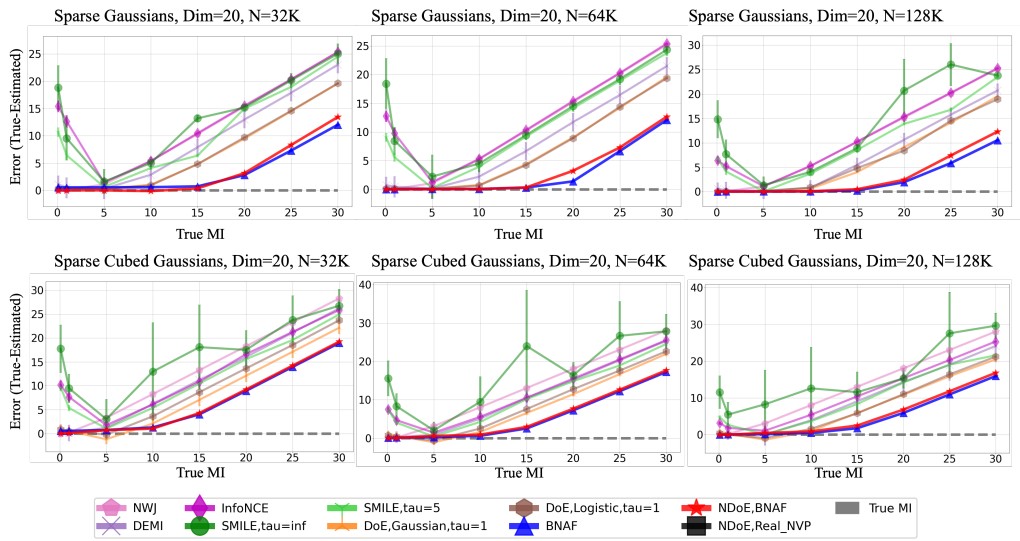

Figure 5: MI estimation between multivariate Sparse Gaussian variables (Top) and between multivariate Sparse Gaussian variables with a cubic transformation (Bottom). The size of training data are 128K. The estimation error $(I(x, y) - \hat{I}(x, y))$ are reported. Closer to zero is better.

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
