## A   REAL NVP

An alternative architecture for implementing our approach is based on Real NVP. Real NVP, proposed by Dinh et al. (2016), is a class of normalizing flows constructed using simple and flexible bijections with efficient computation of the Jacobian determinant. Each transformation in Real NVP is referred to as an affine coupling layer. Given a $d$-dimensional input $x$ and a partition point $d_m < d$, the output $y$ of an affine coupling layer is defined by the following equations:

$$y_{1:d_m} = x_{1:d_m}, \ y_{d_m+1:d} = x_{d_m+1:d} \odot \exp(s(x_{1:d_m})) + t(x_{1:d_m}),$$

where $s$ and $t$ represent the scale and translation functions, mapping $\mathbb{R}^{d_m} \to \mathbb{R}^{d-d_m}$, and $\odot$ denotes the Hadamard (element-wise) product. The Jacobian matrix of this transformation is given by:

$$\begin{bmatrix} \mathbb{I}_{d_m} & 0 \\ \frac{\partial y_{d_m+1:d}}{\partial x_{1:d_m}^T} & \mathrm{diag}(\exp(s(x_{1:d_m}))) \end{bmatrix},$$

where $\mathrm{diag}(\exp(s(x_{1:d_m})))$ is a diagonal matrix whose diagonal elements correspond to the vector $\exp(s(x_{1:d_m}))$. As the Jacobian is triangular, its determinant can be efficiently computed as $\exp(\sum_j s(x_{1:d_m})_j)$. Additionally, since the computation does not depend on the Jacobian of $s$ and $t$, arbitrarily complex functions can be used for $s$ and $t$. A common choice is deep convolutional neural networks with more features in the hidden layers than in the input and output layers.

**Example A.1.** *Consider a function $f\colon \mathbb{R}^4 \to \mathbb{R}^4$, given as a composition $f = f^2 \circ f^1 \circ f^0$ with $f^0, f^1, f^2$. Hence, the corresponding neural network has the form shown in Figure 6.*

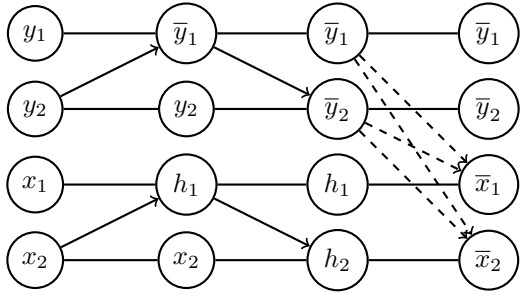

Figure 6: A Real NVP Flow $f(y,x)$. Solid lines represent identical units. Arrows represents the affine coupling transformations.

## B   THEORETICAL RESULTS

The flows we consider to implement our MI estimator are based on the factorization

$$p(x,y) = p(x \mid y) \cdot p(y),$$

with $x \in \mathbb{R}^n$, $y \in \mathbb{R}^n$. The corresponding flows are implemented by block-triangular maps. We call $T\colon \mathbb{R}^n \to \mathbb{R}^n$ triangular, if the following conditions are satisfied for $i \in \{1, \dots, n\}$:

(a) $T_i(x) = g_i(x_1, \dots, x_i)$ (in particular, $\partial T_i / \partial x_j = 0$ for $j > i$);

(b) For any $x_{1:i-1} = (x_1, \dots, x_{i-1})$, $T_i(x_{1:i-1}, x_i)$ is monotonically increasing as a function of $x_i$.

It is easy to see that triangular maps are invertible. It is a well-known result that normalizing flows can be implemented as triangular maps (Bogachev et al., 2007). In our setting, we are interested in block-triangular maps of a specific form. We call a map $T\colon \mathbb{R}^{2n} \to \mathbb{R}^{2n}$ block triangular, if $T = (T_1(y), T_2(y,x))$, where each $T_i\colon \mathbb{R}^n \to \mathbb{R}^n$ and $\det \mathrm{d}T_1(y) > 0$, as well as $\det \mathrm{d}_y T_2(y,x) > 0$ for each $y$ and $x$. While in our case we restrict to the case where $x$ and $y$ have the same dimension, everything that follows generalizes to different dimensions. Recall that if

$T \colon \mathbb{R}^{2n} \to \mathbb{R}^{2n}$ is a map and $\mu$ is a probability measure on $\mathbb{R}^{2n}$, then the push-forward measure is given by

$$T_*\mu(A) = \mu(T^{-1}(A)).$$

Moreover, if $\mu$ has density $p(x, y)$ and we denote by $q$ the density of the push-forward measure, then

$$q(T(y, x)) \cdot |\det \mathrm{d}T(y, x)| = p(y, x).$$

In particular, if $T$ is a normalizing flow, then $q$ is a Gaussian density. In particular, if $T$ is block-triangular as above, then

$$|\det \mathrm{d}T(y, x)| = |\det \mathrm{d}_y T_1(y)| \cdot |\det \mathrm{d}_x T_2(y, x)|$$

and the normalizing flow corresponds to the factorization

$$p(y) = q(T_1(y)) \cdot |\det \mathrm{d}_y T_1(y)|$$
$$p(x \mid y) = q(T_2(y, x)) \cdot |\det \mathrm{d}_x T_2(y, x)|,$$

where again $q$ is a Gaussian density on $\mathbb{R}^n$. It is not a priori clear how this approach leads to a normalizing flow for $p(x)$, say

$$p(x) = q(T_x(x)) \cdot |\det \mathrm{d}_x T_x(x)|,$$

in such a way that $T_2(y, x)$ can be built from this. The following theorem answers this question.

**Theorem B.1.** *Let $\mu, \nu$ be absolutely continuous probability measures on $\mathbb{R}^{2n}$ and let $\mu_1, \mu_2, \nu_1, \nu_2$ denote the marginal distributions on the first and last $n$ coordinate, respectively. Let $T_1 \colon \mathbb{R}^n \to \mathbb{R}^n$, $T_2 \colon \mathbb{R}^n \to \mathbb{R}^n$ be maps such that*

$$(T_1)_*\mu_1 = \nu_1, \quad (T_2)_*\mu_2 = \nu_2.$$

*Then there exists a map $T \colon \mathbb{R}^{2n} \to \mathbb{R}^{2n}$ such that $T_*\mu = \nu$ and*

$$T(y, x) = (T_1(y), \overline{T}(y, T_2(x)))$$

*for a suitable map $\overline{T} \colon \mathbb{R}^{2n} \to \mathbb{R}^n$.*

We note that the existence of $T_1$ and $T_2$ as in the theorem follows from Bogachev et al. (2007) (they may even be chosen to be triangular maps).

*Proof.* Consider the map

$$U \colon \mathbb{R}^{2n} \to \mathbb{R}^{2n}, \quad U(y, x) = (y, T_2(x)).$$

Let $\overline{\mu} = U_*\mu$. By Bogachev et al. (2007), we know that there exists a triangular transformation $V = (T_1, \overline{T}) \colon \mathbb{R}^{2n} \to \mathbb{R}^{2n}$, with $\overline{T} \colon \mathbb{R}^n \to \mathbb{R}^n$ and $V_*\overline{\mu} = \nu$. It follows that

$$\nu = V_*\overline{\mu} = V_*U_*\mu = (V \circ U)_*\mu.$$

Since $V \circ U = T$ as stated in the theorem, the claim follows. $\square$

**Corollary B.2.** *Let $p$ be the joint density of $(X, Y)$ and consider the factorization*

$$p(x, y) = p_Y(y) \cdot p_{X|Y}(x|y).$$

*Let $f^x \colon \mathbb{R}^n \to \mathbb{R}^n$ be a normalizing flow for the marginal density $p_X(x)$. The there exists a block-triangular map $f = (f_1, f_2)$ that is a normalizing flow for $p(x, y)$, where $f_2(y, x) = \overline{f}(y, f^x(x))$ for a map $\overline{f} \colon \mathbb{R}^{2n} \to \mathbb{R}^n$, and such that*

$$p(x|y) = q(\overline{f}(y, f^x(x))) \cdot |\det \mathrm{d}_x f_2(y, x)|,$$

*where $q$ is a Gaussian density.*

The idea is to implement such a flow using neural networks, in such a way that by deactivating a certain part of the neural network for $f_2(y, x) = \overline{f}(y, f^x(x))$, we get the flow $f^x$. In analogy to Block Neural Autoregressive Flows, we now consider

$$f \colon \mathbb{R}^{2n} \to \mathbb{R}^{2n}, \quad f(y, x) = (f_1(y, x), f_2(y, x))^T$$
$$= f^{\ell} \circ \cdots \circ f^1,$$

where $f_1 \colon \mathbb{R}^{2n} \to \mathbb{R}^n$ and $f_2 \colon \mathbb{R}^{2n} \to \mathbb{R}^n$. We assume that each $f^k$, $1 \leq k < \ell$, is of the form

$$f^k(y, x) = \sigma \left( \begin{pmatrix} g(B_{11}^k) & 0 \\ B_{21}^k & g(B_{22}^k) \end{pmatrix} \begin{pmatrix} y \\ x \end{pmatrix} + \begin{pmatrix} b_1^k \\ b_2^k \end{pmatrix} \right)$$

with $B_{ij}^k \in \mathbb{R}^{m_k \times m_{k-1}}$, $g = \exp$ and $m_0 = d$. For $k = \ell$ we omit the activation function and set $m_k = d$. The following basic result is shown along the lines of De Cao et al. (2019).

**Theorem B.3.** *The Jacobian* $\mathrm{d}f$ *is a* $2 \times 2$ *block triangular matrix with* $d \times d$ *blocks,*

$$\mathrm{d}f(y, x) = \begin{pmatrix} \mathrm{d}_y f_1(y) & 0 \\ \mathrm{d}_y f_2(y, x) & \mathrm{d}_x f_2(y, x) \end{pmatrix}.$$

*Moreover, if* $\sigma$ *is strictly increasing, then* $\det \mathrm{d}_y f_1(y) > 0$ *and* $\det \mathrm{d}_x f_2(y, x) > 0$ *for all* $(y, x)$.

One consequence of this characterization of block-triangular flows is that if we train this neural network with the cost function

$$\log q(f_2(y, x)) + \det \mathrm{d}_x f_2(y, x),$$

we can obtain the entropy associated to the marginal density, $H(X)$, by "deactivating" the weights that operate on $y$. Corollary B.2 suggests that given enough expressive power of our neural network architecture, we can train the network to both approximate $H(X|Y)$ and $H(X)$ by first training the marginal density $f^x$ and then training for $(f_1, \bar{f})$ on samples $(y_i, f^x(x_i))$. Along the lines of the appendix in De Cao et al. (2019), one can show that any conditional density can be approximated by a block-triangular flow as described.

## C    ADDITIONAL EXPERIMENTS

### C.1    MI ESTIMATION WITH NONLINEAR TRANSFORMATIONS

The first experiment we conducted are focused on estimating MI with samples that are generated with nonlinearly transformed Gaussians. Here we consider asinh and wiggly transformations that are provided in (Czyż et al., 2023). The different MI estimators were trained on the 20-dimensional datasets of the varying sample size (32K, 64K and 128K). All the other settings remained the same.

**Results.** The results are presented in Figure 7 and Figure 8. Overall, all discriminative methods tend to underestimate MI, while our proposed methods demonstrated superior performance. In particular, **NDoE, BNAF** exhibited less bias compared to **BNAF** in cases with additional cubic transformations. Although **NDoE, Real NVP** performed worse than both **NDoE, BNAF** and **BNAF**, it still exhibited less bias than all other discriminative methods and the **DoE** estimators across all scenarios.

### C.2    MI ESTIMATION ON EXTREMELY SMALL-SIZED SAMPLES

The second experiment we conducted is similar to the last experiment in Section C.1. We trained the different estimators on the training set of size 1024 and tested them on another independently generated 1024 samples to obtain MI estimates. All the other settings remained the same. We repeated the training process for $20, 50, 100$ and $200$ epochs.

**Results.** The results of 20 dimensionalities are presented in Figure 9. With the small number of epochs of training on extremely small-sized samples, all methods gave a bad performance on large MI. However, **DEMI** and **NDoE, BNAF** still obtained relatively good estimates when the true MI is close to zero. With the increase of the number of training epochs, the estimates of **NDoE, BNAF** started to converge to the true MI, while other discriminative methods lead to large errors. We noticed that **BNAF** also shows the trend of convergence, but the estimation results are worse than **NDoE, BNAF**. **NDoE, Real NVP** failed to achieve realistic results in this experiment.

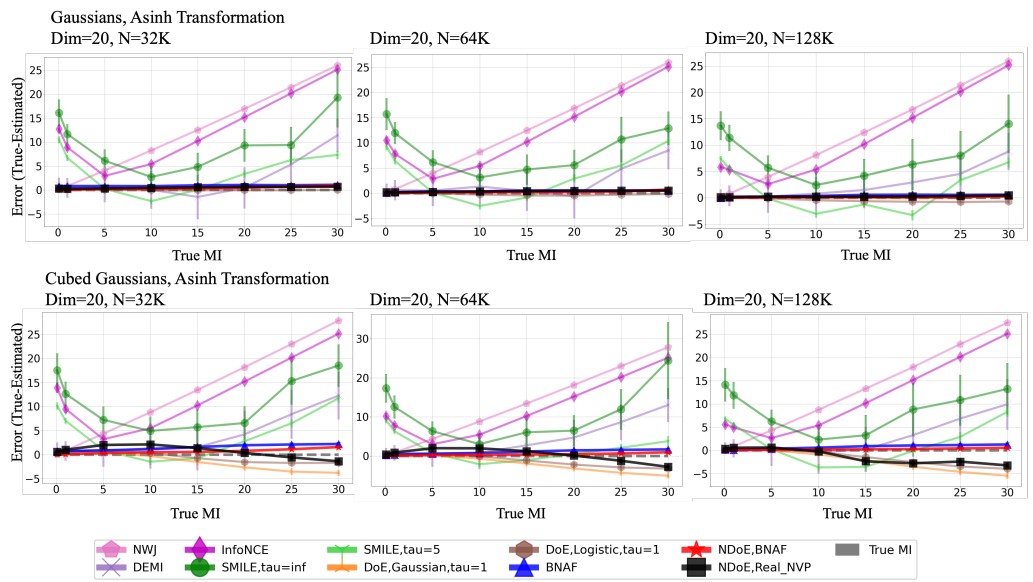

Figure 7: MI estimation between asinh-transformed Gaussian variables (Top) and between asinh-transformed Gaussian variables with a cubic transformation (Bottom). The estimation error $(I(x,y) - \hat{I}(x,y))$ are reported. Closer to zero is better.

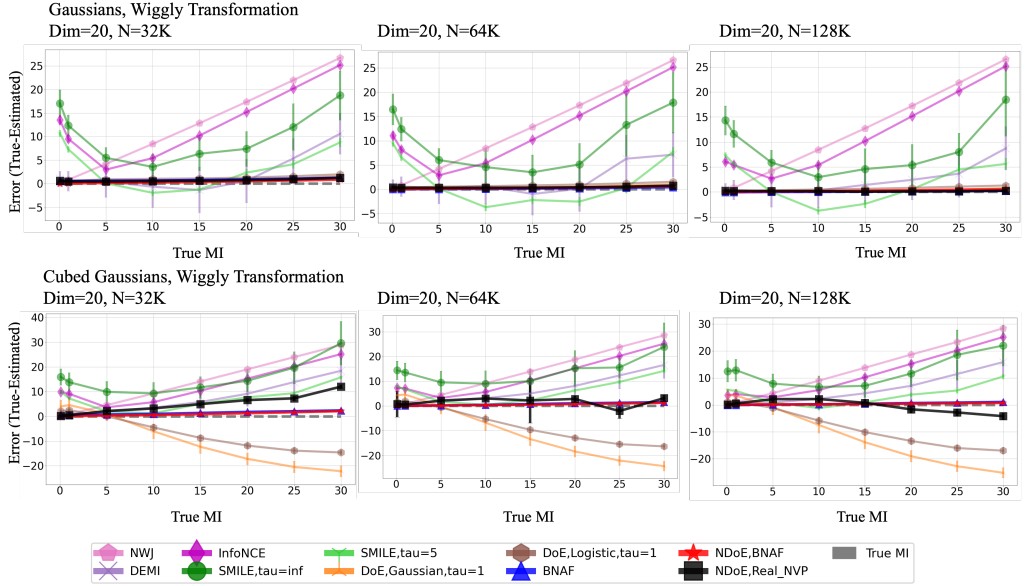

Figure 8: MI estimation between wiggly-transformed Gaussian variables (Top) and between wiggly-transformed Gaussian variables with a cubic transformation (Bottom). The estimation error $(I(x,y) - \hat{I}(x,y))$ are reported. Closer to zero is better.

### C.3    MI ESTIMATION ON CORRELATED UNIFORMS AND STUDENT'S T DISTRIBUTIONS

It is well-known that non-Gaussian distributions, especially distributions with long tails, remains challenging for MI estimation for many reasons. In this experiment, we sampled from two random variables $X$ and $Y$ of correlated Uniform and Student's t distributions, for which the MI can be exactly obtained from their known correlation. The construction of Student's t distribution follows the idea of Czyż et al. (2023) where the Gaussians are taken from the first experiment. Note that,

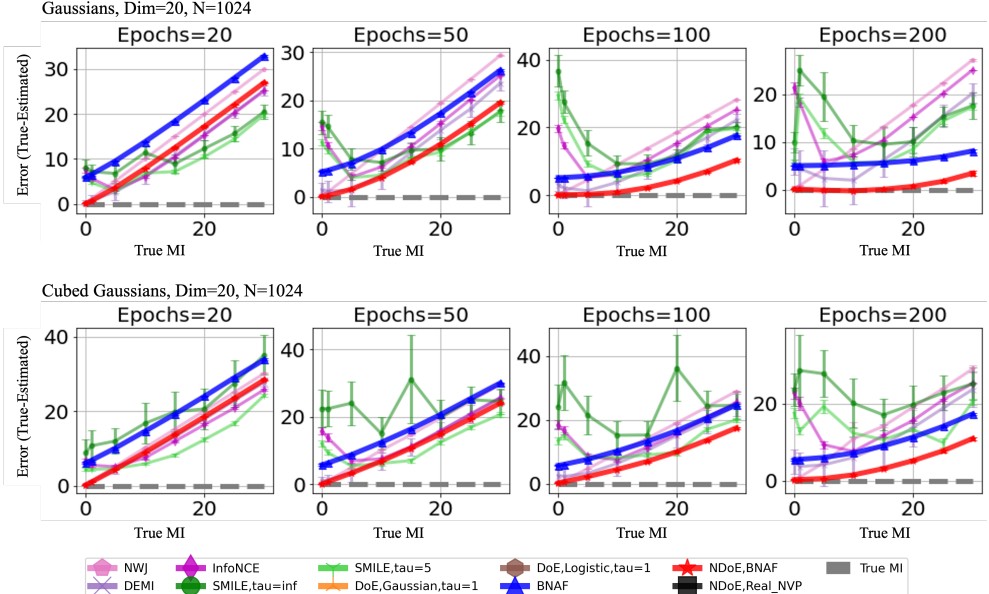

Figure 9: MI estimation between multivariate Gaussian variables (Top) and between multivariate Gaussian variables with a cubic transformation (Bottom). The estimators are trained on training data of size 1024 with varying training epochs and the estimates are obtained from testing data of size 1024. The estimation error $(I(x,y) - \hat{I}(x,y))$ are reported. Closer to zero is better.

$I(X,Y) > 0$ for the generated Student's t distribution even for independent Gaussians $X, Y$ in this example. We trained the different estimators on the training set of 128K samples and tested them on another independently generated 10240 samples to obtain MI estimates. All the other settings remained the same.

**Results.** The results for Uniform variables are shown in Figure 10, while those for Student's t variables are presented in Figure 11. In the Uniform case, our proposed method provides better estimates with relatively small variance. **DEMI** achieves competitive results for small MI, but the error remains substantial for larger MI values. Notably, **NDoE, Real NVP** delivers even better performance when a cubic transformation is applied.

Unfortunately, most estimators fail to yield realistic results for Student's t distributions, whereas our method maintains small bias and variance in the estimates. **InfoNCE** displays a large bias for high MI, and **DEMI** shows very high variance. This experiment also excludes the influence of learning the target distribution from the same base distribution (Simple Gaussian) in **NDoE** and **BNAF**. Although the bias increases compared to the Gaussian examples, **NDoE** continues to demonstrate superior performance compared to other methods.

### C.4 COMPARISON BETWEEN NDoE AND BNAF

To demonstrate the empirical out-performance of our **NDoE, BNAF** method against the core baseline, **BNAF**, which utilizes two separate flows with identical hyperparameters and initializations, we consider the cubed Gaussians distribution as an example. The estimated MI is plotted against the number of training epochs to showcase the comparative performance. The results are illustrated in Figure 12. All other settings remain the same.

**Results.** As shown in the plots, **NDoE, BNAF** demonstrated better convergence behavior with relatively small error after just 10 epochs of training, whereas **BNAF** required between 20 to 60 epochs to achieve competitive results. This performance gap is particularly pronounced when the number of training samples is smaller. Another noteworthy observation is that **NDoE, BNAF** empirically behaved as an upper-bound estimator, consistently producing estimates greater than zero, which aligns well with the fundamental properties of mutual information (MI). In contrast, **BNAF**

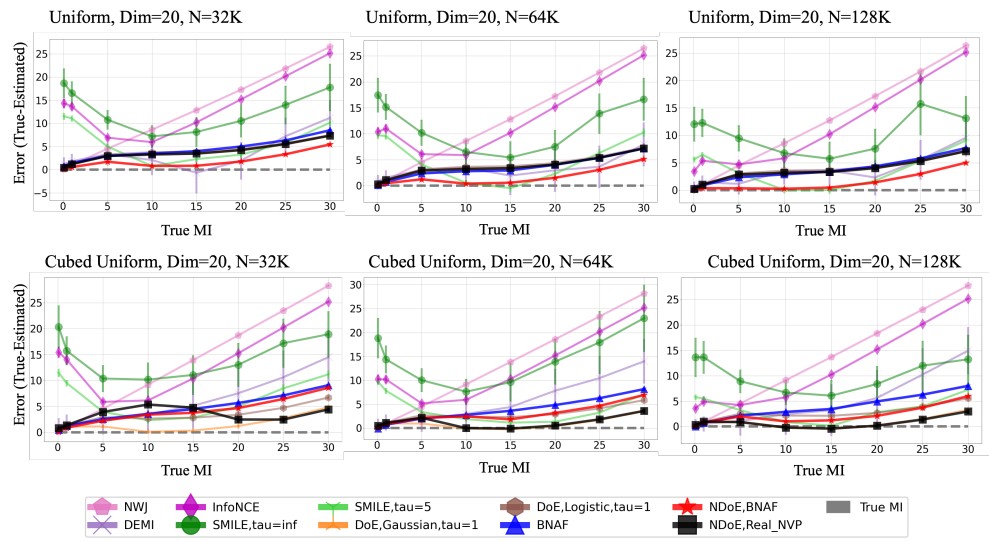

Figure 10: MI estimation between multivariate Uniform variables (Top) and between multivariate Uniform variables with a cubic transformation (Bottom). The estimation error $(I(x, y) - \hat{I}(x, y))$ are reported. Closer to zero is better.

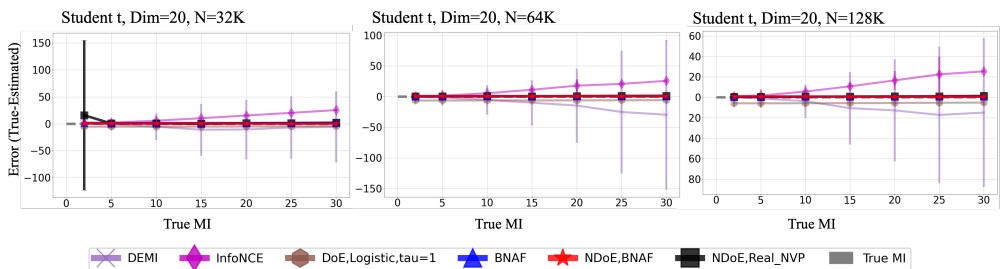

Figure 11: MI estimation between multivariate Student's t random variables. The estimation error $(I(x, y) - \hat{I}(x, y))$ are reported. Closer to zero is better.

does not exhibit this property, which is often regarded as a key limitation of generative methods. We attribute this phenomenon to the Correlation Boosting Effect proposed by Gao et al. (2017), though we do not provide a rigorous proof at this stage.

## C.5 LONG-RUN TRAINING BEHAVIOR OF NDOE

We noticed from the last three experiments that **DEMI** underestimates the MI when the random variables are highly dependent. The underestimation is not alleviated by the increased number of training epochs. At the same time, our method shows the reduction of bias with the repetition of training. Thus, we proposed an assumption that the discriminative methods diverge to the true MI for high mutual dependence, while generative models have a good convergence property with sufficient training samples, with the precondition of enough expressive power of the neural network. In other words, there is a systematic bias in the discriminative methods which are positively correlated to mutual dependence. To verify this experimentally, we conducted a long-run training behavior experiment using similar settings of Liao et al. (2020). Here we only verify the long-run training behavior for **NDoE, BNAF**. All the estimators were trained on the 20-dimensional Gaussian and Cubed Gaussian case for 100000 training steps with batch size 128. Samples were drawn directly from the generating distributions. We did this for four ground-truth MI values of $0.1, 10, 20,$ and $30$.

**Results.** The results are shown in Figure 13 and Figure 14. Among all the methods, our method shows the best convergence behaviour for all MI with the increase of training epochs, and the variances

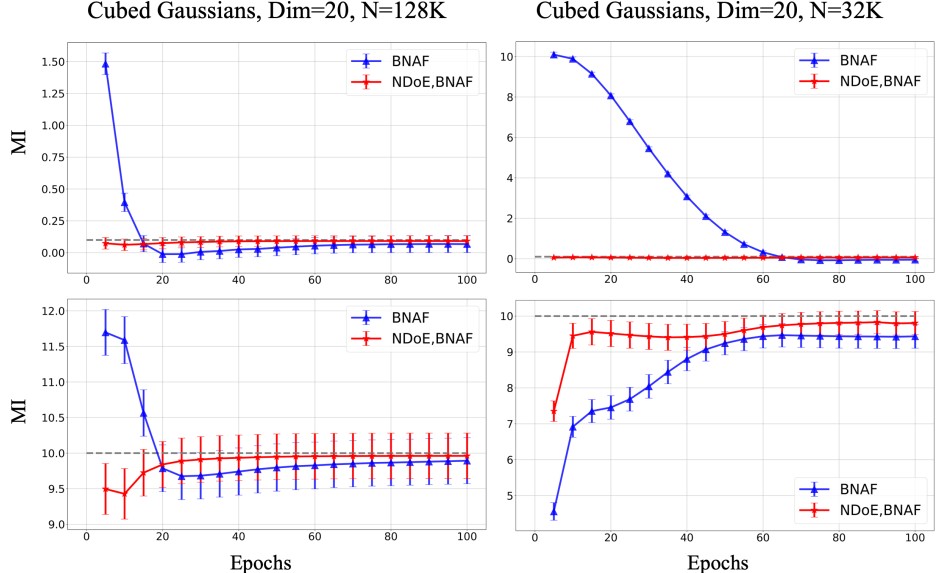

Figure 12: MI estimation between multivariate cubed Gaussian variables. The estimation $\hat{I}(x,y)$ of varying training epochs versus the true underlying MI are reported.

remains relatively small **DEMI** method is competitive when true MI is close to $0$. However, it underestimates MI vastly for large true MI. We also noticed that **SMILE** with the parameter $\tau = 1, 5$ tends to overestimate MI even though they are based on a lower bound of it. McAllester & Stratos (2018) suggests that the reasoning behind this could be the sensitivity of the estimate of $-\ln \mathbb{E}[e^{f(x,y)}]$ to outliers. However, the parameter $\tau$ actually clips the term $e^{f(x,y)}$ to the interval $[e^{-\tau}, e^{\tau}]$, which removes outliers from the neural network outputs.

### C.6 ASYMMETRY TEST AND VARYING BATCHSIZE

It is obvious to see that our proposed method is asymmetric, i.e. estimating the MI by using $I(X,Y) = H(X) - H(X,Y)$ and $I(X,Y) = H(Y) - H(Y,X)$ could obtain different results. We did an extra experiment on the above distribution to show that the difference is minor. The results are shown in Figure 15. Since our method tends to underestimate the true MI in the experiments, choosing the larger estimation will lend to less bias in most cases. Another experiment focuses on varying batchsize. It is believed that the poor performance of discriminative methods on high MI estimation are dependent on the batchsize. We choose the batchsize of $64, 128, 256$ and $512$ to see the performance of our method and compare them with the baselines. The results are shown in Figure 16.

## D SELF-CONSISTENCY

In applications with real data, obtaining the ground truth MI is challenging or not possible. However, as suggested by Song & Ermon (2019), one can still test whether a MI estimator satisfies some of the fundamental properties of MI: $I(X,Y) = 0$ if $X$ and $Y$ are independent, the data processing inequality is satisfied (that is, transforming $X$ and $Y$ should not increase the MI), and additivity.

Following Song & Ermon (2019), we conducted self-consistency tests on high-dimensional images (MNIST) under three settings, where obtaining the ground truth MI is challenging. These settings involve processing images $X$ and $Y$ in different ways to assess the performance of various methods: **DEMI**, **InfoNCE**, **SMILE**, **NDoE, BNAF** and **BNAF**, where **NDoE, BNAF** and **BNAF** applies autoencoders (AE) for dimensionality reduction. **DoE** is not included as it is considered for certain failure in the experiments before. The three settings include:

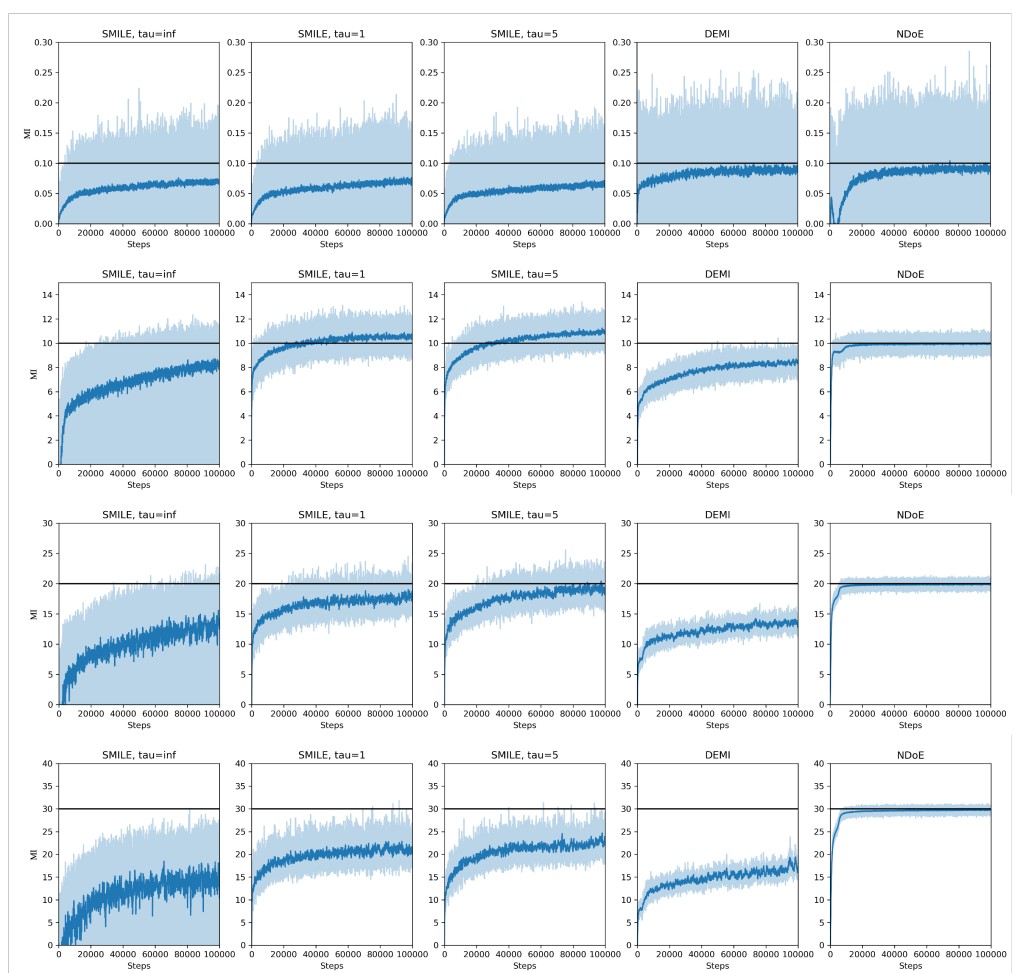

Figure 13: MI estimation between multivariate Gaussian variables. The estimation of each training step $\hat{I}(x,y)$ versus the true underlying MI are reported. The estimators are trained on 100000 training steps with varying true MI.

(a) $X$ is an image, and $Y$ is the same image with the bottom rows masked, leaving the top $t$ rows. The goal is to observe whether MI is non-decreasing with $t$. Methods are evaluated under various $t$ values, normalized by the estimated MI between $X$ and itself.

(b) Data-Processing. $X$ corresponds to two identical images, and $Y$ comprises the top $t_1$ and $t_2$ rows of the two images ($t_1 \geq t_2$). The evaluation involves comparing the estimated MI ratio between $[X, X]$ and $[Y, h(Y)]$ to the true MI between $X$ and $Y$, where $h(Y)$ use $t_2 = t_1 - 3$ rows.

(c) Additivity. $X$ corresponds to two independent images, and $Y$ includes the top t rows of both. The assessment focuses on the estimated MI ratio between $[X_1, X_2]$ and $[Y_1, Y_2]$ relative to the true MI between $X$ and $Y$.

**Results.** The results are shown in Figures 17. Regarding the baseline, most methods correctly predict zero MI when $X$ and $Y$ are independent, thereby passing the initial self-consistency test. Additionally, the estimated MI shows a non-decreasing trend with increasing $t$, although the slopes differ among the methods. The ratio obtained by **NDoE, BNAF** is very close to the true ratio.

For the data-processing test, we set $t_2 = t_1 - 3$. Ideally, the estimator should satisfy $\hat{I}([X, X]; [Y, h(Y)])/\hat{I}(X, Y) \approx 1$. This is because additional processing should not result in an increase in information. All methods performs relatively well except for **NDoE, BNAF** and **BNAF**. This is possibly due to limited capacity of AE.

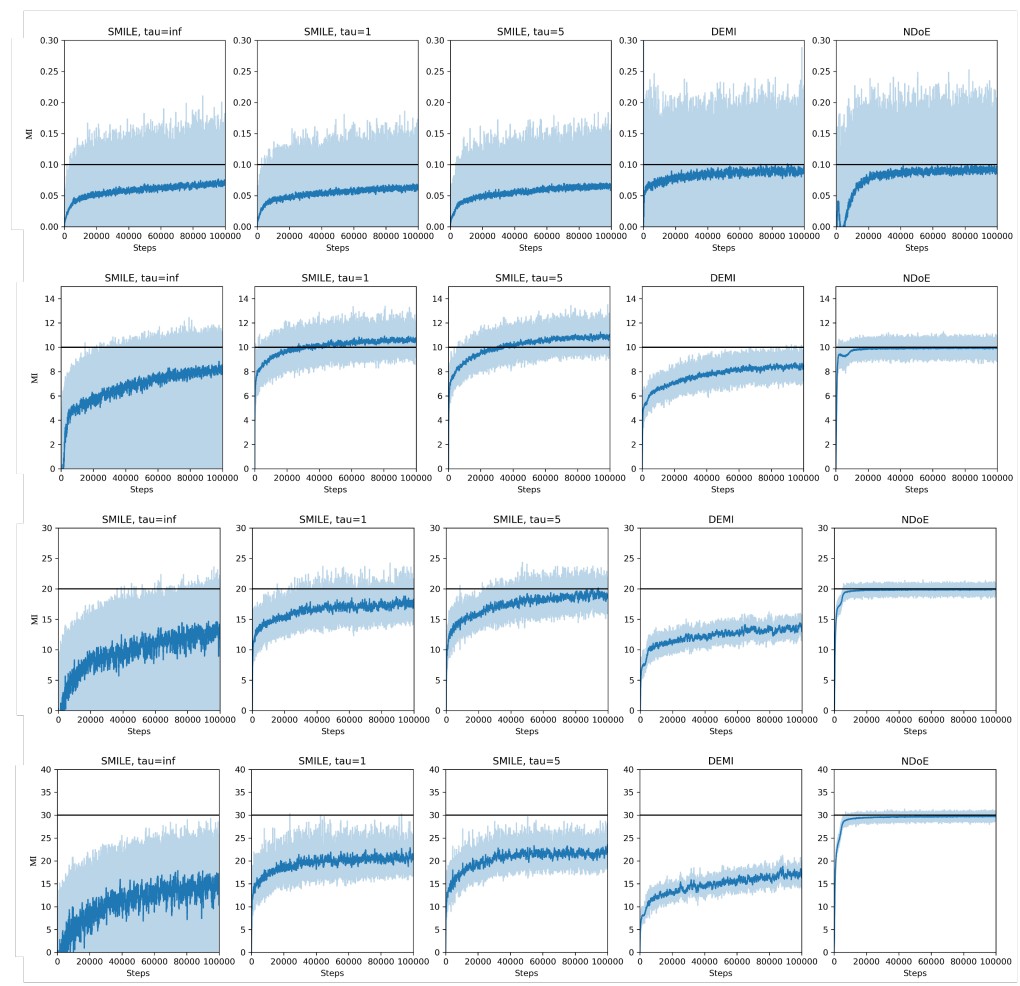

Figure 14: MI estimation between multivariate Gaussian variables with a cubic transformation. The estimation of each training step $\hat{I}(x, y)$ versus the true underlying MI are reported. The estimators are trained on 100000 training steps with varying true MI..

In the additivity setting, the estimator should ideally double its value compared to the baseline with the same t, i.e. $\hat{I}([X_1, X_2]; [Y_1, Y_2])/\hat{I}(X, Y) \approx 2$. Discriminative approaches did not perform well in this case, except when t was very small. As t increased, this ratio converged to 1, possibly due to initialization and saturation of the training objective. However, **NDoE, BNAF** performed well on this test except when t is small ($t = 0, 3$). Compared with the results from Song & Ermon (2019), it is promising to see improving performance by using VAE instead.

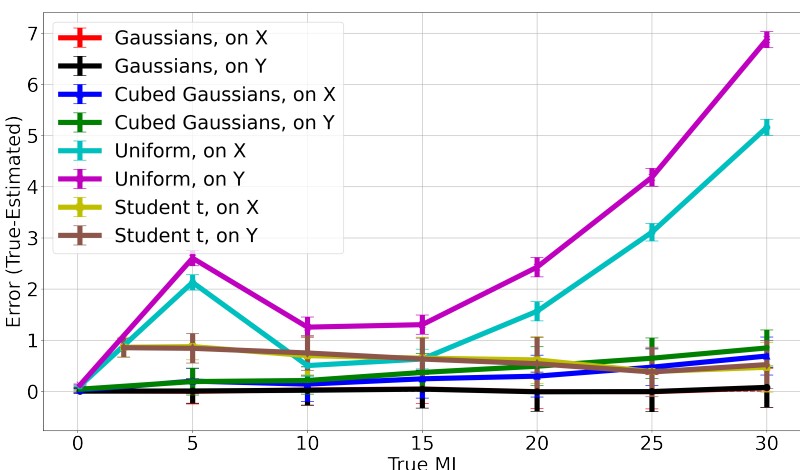

Figure 15: MI estimation between random variables of different distributions. The estimation error $(I(x,y) - \hat{I}(x,y))$ are reported. Closer to zero is better.

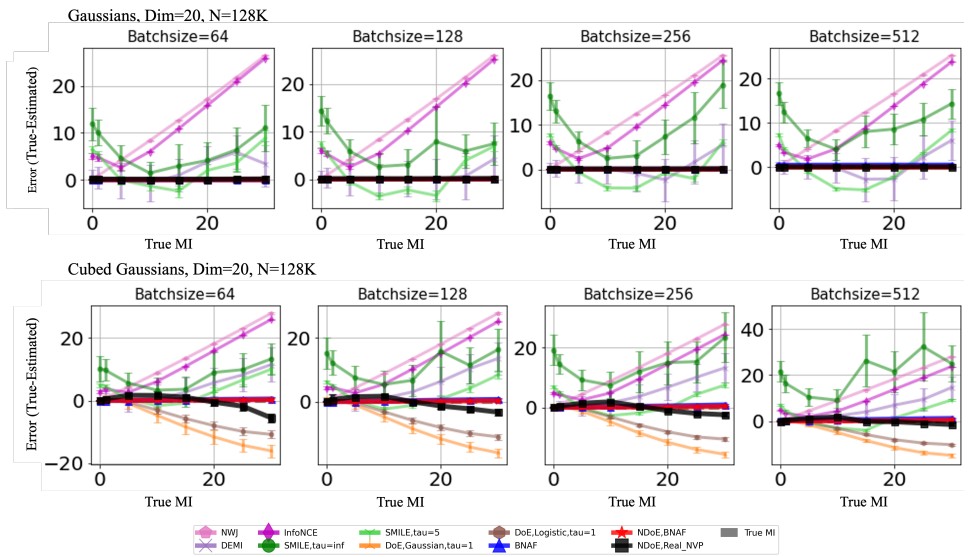

Figure 16: MI estimation between multivariate Gaussian variables (Top) and between multivariate Gaussian variables with a cubic transformation (Bottom). The estimators are trained with varying training batchsize. The estimation error $(I(x,y) - \hat{I}(x,y))$ are reported. Closer to zero is better.

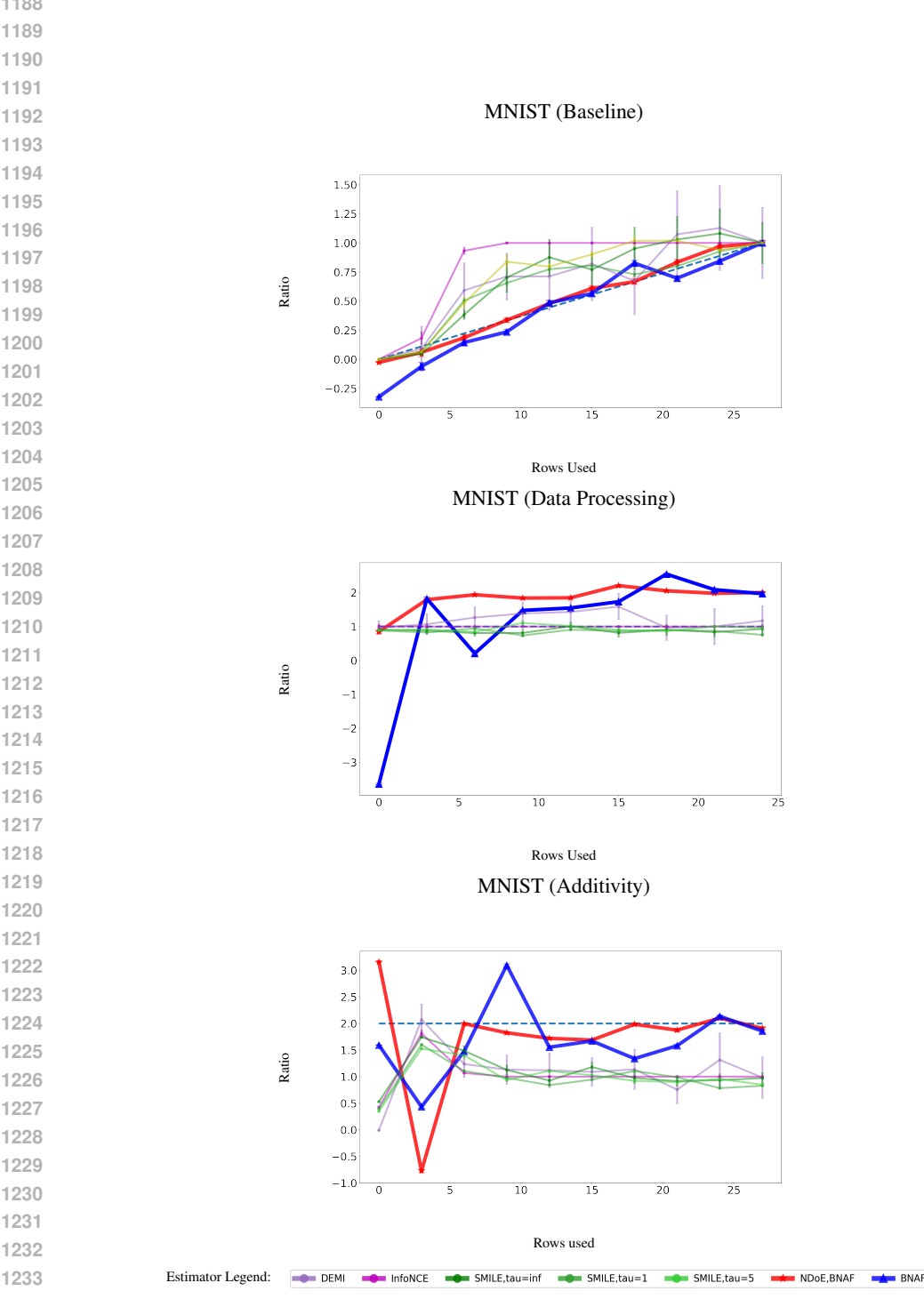

Figure 17: Evaluation on high-dimensional images (MNIST) under three settings. From top to bottom: Evaluation of $\hat{I}(X;Y)/\hat{I}(X;X)$; Evaluation of $\hat{I}([X,X];[Y,h(Y)])/\hat{I}(X;Y)$, where the ideal value is 1; Evaluation of $\hat{I}([X_1,X_2];[Y_1,Y_2]/\hat{I}(X;Y)$, where the ideal value is 1. $X$ is an image, $Y$ contains the top $t$ rows of $X$ and $h(Y)$ contains the top $(t-3)$ rows of $X$.