# OpenReview forum: "A Normalizing Flows based Difference-of-Entropies Estimator for Mutual Information"
_ICLR.cc/2025/Conference — Submitted to ICLR 2025_

### Official Review · Reviewer_KwGE · 2024-10-25

**Soundness:** 3
**Presentation:** 3
**Contribution:** 3
**Rating:** 5
**Confidence:** 3

**Summary:**

The paper proposes a new difference of entropies estimator for mutual information (MI). The estimator uses a block autoregressive flow and shows good performance on benchmarks.

**Strengths:**

- Good summary of exisiting methods
- Clearly written
- Promising performance on benchmarks

**Weaknesses:**

- Robustness of results to different hyperparameter settings could be explored in more depth
- Figures are hard to read and not colorblind friendly
- Minor: Reference section deserves revision; many inconsistencies

**Questions:**

- How do methods compare when using half or double the parameter count?
- Have you considered alternative ways of reporting the results? The plots feel very crowded and are hard to discern.

---

> ### Author Response · Authors · 2024-11-27
>
> We improved the presentation of the figures and the discussion in the appendix in the revision. We are open to suggestions for alternative ways of presenting the results in follow-up work.

---

> > ### Comment · Reviewer_KwGE · 2024-11-29
> >
> > Thank you for updating the figures, making them colorblind friendly. I agree with reviewer vKnh that they still feel a little crowded though. My other points have not been addressed and I maintain my score.

---

### Official Review · Reviewer_MQzj · 2024-10-27

**Soundness:** 4
**Presentation:** 4
**Contribution:** 3
**Rating:** 8
**Confidence:** 4

**Summary:**

This work presents a novel method for the estimation of mutual information between arbitrary, continuous random variables, extending the range of estimators from the literature. The key idea stems from the seminal work by McAllester & Stratos (2018), in which mutual information is cast as an optimization problem by noting that entropy and conditional entropy can be estimated as infimum of cross-entropies computed with an approximation density.

Such density, in the presented work, is obtained by the application of normalizing flows, that transform a base normal distribution into an arbitrary distribution, which is used in the optimization problem discussed above. The authors clearly indicate how to use recent advances in the theory and practice of normalizing flow to build an approach that is computationally cheap, by means of an amortization approach. Indeed, they can estimate mutual information, and the elements described above (entropy and conditional entropy) with a single network.

A large number of synthetic experiments complement the presented methodology, illustrating the advantages of the proposed method when compared to a number of alternatives from the literature. Such experiments include cases in which the original Gaussian distributions are transformed by means of non-linear functions. Furthermore, the appendix contains additional experiments including self-consistency tests, as done previously in the literature.

**Strengths:**

* Section 2.1 is very clear, and contains a good summary of known results from McAllester & Stratos (2018) in sections 5.1 and 5.2, setting the stage for a much advanced parametrization of the two optimization problems of estimating cross entropies to obtain estimates of entropy and conditional entropy, by means of an amortized, flow-based approach. I think this is an original idea which displays good performance in practice.

* Section 2.2 is also very clear and didactical, motivating the need for efficient variants of the base normalizing flow approach, which can be costly from the computational perspective. Ultimately, a block autoregressive formulation is what the authors used for MI estimation. I really like, again, the clear and didactical approach to explain the “amortization technique” to activate part of the flow network to obtain estimates of the various quantities required to solve the optimization problem in Equation 3.

* Despite some questions (see below), the experimental section is very thorough (including also the results presented in Appendix C, which complement substantially the standard benchmark results in the main part of the paper).

**Weaknesses:**

* Expression at line 37 is the equivalent of equation 7 in McAllester & Stratos (2018), which is a difference of entropies. The authors anticipate equation 14 in McAllester & Stratos (2018), which instead involves cross-entropies as upper bounds of the entropy and conditional entropy. I think this can be easily clarified in the paper, as the authors correctly characterize their expressions as the infimum of cross entropies that they translate to a variational problem.

* Section 1.1 offers a detailed overview of alternative MI estimators, but misses one important approach that has appeared prior to the last reviewed method from Butakov et al (2024), namely the MINDE estimator proposed in Franzese et al., “Mutual information neural diffusion estimator”, ICLR 2024, which also targets arbitrary, high-dimensional continuous distributions, making it a good candidate for comparison in the experimental evaluation.

* Experiments in section 4 rely only on synthetic data, which is a necessity to gain access to ground-truth MI, and to perform a comparative analysis among methods. The authors build upon prior benchmark studies, and propose a series of synthetic random variables sampled from Gaussian distributions with varying dimensionality, and having access to various sample sizes. They also consider one non-linear transformation by applying a cubic function to one of the variables. While in all such cases, the proposed method performs well, I am curious to understand why (also by looking at experiments in Appendix C, including additional transformations) the proposed method struggles with highly non-linear transformations. If on the one hand, the authors claim that the superiority of the proposed method in the Gaussian case might be “likely be due to the fact that the base distribution is itself Gaussian” (line 409), when this is not the case, does it mean that normalizing flows struggle with arbitrary distributions? This should not make sense right? So what is the problem, which is exacerbated by an high MI regime?

* One last question on the experiments is in order. Recent work, such as Kong et al, “Interpretable Diffusion via Information Decomposition”, ICLR 2024, Franzese et al. “Mutual information neural diffusion estimator”, ICLR 2024, illustrate some practical applications in which mutual information estimation can be instrumental. Have the authors attempted at estimating MI between complex distributions such as $X \sim \text{image data}$ and $Y \sim \text{Text embeddings}$? This question is important to fully grasp the potential impact of MI estimators that can be useful in the machine learning community for a variety of purposes.

**Questions:**

I do not have more questions than those discussed in the weaknesses. Apart from missing one reference that I saw recently, and some questions on the experiments, this is a nice work, in my opinion.

---

> ### Author Response · Authors · 2024-11-27
>
> We would like to thank the reviewer for the thoughtful and detailed report.
>
> * On line 37, this is a mistake: it is supposed to read ''conditional entropy'' instead of ''cross-entropy''. Good catch!
>
> * We added a reference to MINDE, but due to time constrained could not include comparisons.
>
> * Using separate base distributions would indeed be an interesting route to explore, we mention this in the conclusions.
>
> * This is an excellent point. We added a line on this in the conclusions section, as a suggestion for future / follow-up work, as there is not enough time and space to do this justice in the current version of the paper.

---

### Official Review · Reviewer_vKnh · 2024-10-28

**Soundness:** 3
**Presentation:** 2
**Contribution:** 2
**Rating:** 5
**Confidence:** 4

**Summary:**

The authors propose using a version of DoE estimators to create an unbiased version of parameterizing normalizing flows to estimate mutual information. They do this by deactivating certain parts of the network to estimate each of the two entropy terms. They demonstrate their method across a variety of synthetic distributions with different transformations.

**Strengths:**

The current method can be flexibly parameterized any type of normalizing flows with a conditional dependence for MI estimation. Theory wise, everything is quite clear and intuitive.

**Weaknesses:**

Currently, the experiments test against a lot of baselines, but don't particularly highlight their main contribution in terms of the addition of the DoE estimator (given that a normalizing flows for MI estimation paper exists). The main comparison experimentally is with standard BNAF, and the results for that aren't fully convincing of the advantages of the method.
One simple way to add an additional comparison is to have a standard RealNVP without the NDoE portion, for an apt comparison. Alternatively, adding DoE to some of the other baselines presented (while cutting down on the total number, as the error bars are quite hard to read with that many baselines, many of which do not contribute particularly to the argument) would also be good. Also, you may want to consider presenting some of the baselines in a table instead (perhaps in the appendix).
The abstract doesn't seem to contain your main contribution here, which can be quite confusing.

**Questions:**

Some of the charts seem to be missing a baseline (e.g. NDoE, RealNVP in Figure 5 and Figure 9), which I presume is due to the line in the text where it is stated that NDoE, Real NVP failed to achieve realistic results in the Sparse Gaussian case. Is this something due to the RealNVP, or did the NDoE part also affect it? Is there some intuition or explanation for why this happened?
Is there a reason for not including a standard RealNVP approach (without the NDoe) in the baselines?

---

> ### Author Response · Authors · 2024-11-27
>
> We appreciate the feedback on the experiments that helps improve the submission. As the reviewer suggested, we cut down on the number of baselines and enhanced the visual presentation of our plots. To further highlight the strength of our method, we add one more experiment in Appendix C.4.
>
> Experimentally speaking, although estimating via separate flows (BNAF) and our method will eventually converge to true MI with enough expressive power and careful training according to universal approximation theory, our method has early convergence especially when samplesize is small. Another feature of our method is that its estimation has constraints of $I(X,Y)>=0$ and it has the similar behavior of a upper bound estimator. This provide a good estimation of small true MI, which is considered a limitation of generative models in this area before. We believe that this may be related to the correlation boosting effect proposed by (Gao et. al, 2018).
>
> Question: NDoE, RealNVP fails to provide estimations for Sparse Gaussian case, which might due to the structure of RealNVP. RealNVP is similar to coupling layers which may have less expressive power in learning sparse interactions of variables. However, we only implement a simple version as a flavor of image-friendly normalizing flows. It should work better with more sophisticated and hierarchical models. The NDoE part does not affect it since the comparison between standard RealNVP and NDoE, RealNVP is very similar to the comparison between standard BNAF and NDoE, BNAF.  We choose not to include it only because we wanted to focus on NDoE, BNAF primarily and NDoE, RealNVP is just an add-on. We do not want to confuse readers with multiple baselines that behave almost the same.

---

> ### Comment · Reviewer_vKnh · 2024-11-28
>
> Thanks for providing some additional experiments an explanations. A couple of questions:
>
> 1.  Figure 12 has 4 plots but only two labels; I'm having a hard time figuring out what each of the plots is specifically trying to designate since the top and bottom entries of the rows have wildly varying results. Also read the corresponding section in the text but not gleaning anything from there.
>
> 2. I'm not very convinced by the claim that the constraint $I(X,Y) \geq 0$ is imposed. (On another related note, in Figure 12 the MI difference seems to be negative? Not sure if this is due to X and Y being switched in the calculation or something else) The standard error bars of the NDoE BNAF seems to extend across 0 by quite a bit, which I'm having a hard time believing that for all 10 setups it achieves a consistent lower upper bound.
>
> Also, figures are still a little bit crowded in some of the main sections in my opinion, particularly due to the error bars.

---

> > ### Author Response · Authors · 2024-12-01
> >
> > 1. We are sorry for the confusion and will clarify this in a revised version. The top row is for true MI is 0.1 and the bottom row is for true MI is 10, so we can look into both small and large MI.
> >
> > 2. By having the estimation for $p(x)$ with a normalizing flow and then applying conditional NNs/adapters for learning $p(x|y)$ on the same flow, it is quite intuitive that the obtained $p(x|y)$ will be smaller than $p(x)$. This is because it acts as a reduction of uncertainty on $x$. This may not be true for two separate flows as they learn $p(x)$ and $p(x|y)$ as different densities.  It can never be perfect aligned in the results due to bias introduced, especially when true MI is close to zero, but it still benefits from such behaviour. In fact, almost all variational bounds based estimators (typical example is MINE) are not bounds but just estimates of MI. See Section 2.2 in B. Poole, 2019.
> >
> > Due to the time limit we have not worked on the figures’ layout, but we assure you that this will be improved before making the paper public.

---

### Official Review · Reviewer_USyt · 2024-11-03

**Soundness:** 3
**Presentation:** 3
**Contribution:** 2
**Rating:** 5
**Confidence:** 4

**Summary:**

This paper expands on the difference-of-entropies (DoE) mutual information (MI) estimator
proposed by David McAllester and Karl Stratos in 2018.
In contrast to the original work, the authors use normalizing flows to model $p(X)$ and $p(X \mid Y)$,
which makes the estimator consistent.
Additionally, a clever trick is proposed to enable the estimation of $p(X)$ and $p(X \mid Y)$ via a single model.
This is achieved through the usage of the block autoregressive flows.
The paper includes comprehensive experimental results that highlight the practical advantages and disadvantages of the approach.

**Strengths:**

1. The paper is overall well-written, the motivation behind the DoE estimator and autoregressive models is presented clearly.
1. The idea of modelling two distributions via one flow model is interesting
   and is reminiscent to a similar technique successfully used in MINDE [1].
1. The scale of the final comparison (the number of other NN-based estimators featured) is truly commendable.
   This allows for a better assessment of the advantages of the proposed approach.
   The authors also provide a comprehensive analysis of the results.
1. Overall, the proposed method achieves better results compared to other NN-based approaches.

**Weaknesses:**

1. Despite the strength №3,
   the set of benchmarks used to evaluate the estimators is very limited and can be considered outdated.
   The authors employ some simple tests from (Czyż et al., 2023),
   but do not consider the distributions which might pose a real challenge to flow-based approaches due to the manifold-like structure:
   the Swiss Roll embedding and the spiral diffeomorphism.
   Additionally, in (Butakov et al., 2024) and in [2], several complex and high-dimensional image-like datasets
   with tractable MI have been proposed.
   Although the authors conduct a number of tests on the MNIST dataset, checking that selected properties of MI also hold for their estimator,
   the work would still benefit greatly from image-like tests for which the ground truth value of MI is available.
1. Although I clearly see that the DoE estimator combined with two expressive enough flow models is consistent,
   a rigorous proof still has to be provided in order to show that the same holds for using only one model;
   please, see the questions.
1. Combining the proposed estimator with a dimensionality reduction technique during the tests with MNIST seems unfair.
   If the proposed estimator fails to estimate MI between images
   (which definitely might happen due to certain limitations of the generative models used),
   this should be clearly represented as a limitation of the method.
1. The major limitation of the proposed method in its current form is that it is only applicable to continuous distributions,
   whereas critic-based methods (MINE, InfoNCE, ...) work with any types of distributions out-of-the-box.
   The authors should address this limitation properly in their manuscript.
   I also suggest dedicating a separate paragraph to all the limitations of the proposed method.

**Minor:**

1. The novelty of this work is limited due to the main ideas behind the DoE estimator being explored in (McAllester & Stratos, 2018).
1. The authors do not compare their method to other approaches based on generative models,
   such as [1] and (Ao & Li, 2022; Duong & Nguyen, 2023; Butakov et al., 2024).
1. The first plot in Figure 16 features a dashed line, which is misleading.
   For this particular test, there are no clues which ratio we should expect to see,
   as the information about $X$ can be distributed non-uniformly among the rows.
   Moreover, the test itself is ill-posed, as $I(X;X) = I(X;Y) = +\infty$ in this particular case;
   I, however, acknowledge that the test is borrowed from the work of Song & Ermon (2020).
1. Due to the source code being absent from the supplementary materials, the reproducibility can be questioned.

**Questions:**

1. As only $p(X)$ and $p(X \mid Y)$ are modeled,
   why can not we always use the original values of $y$ to condition the corresponding flows for $x$?
   Is there any need to assume continuity of $Y$ and apply transformations to $Y$ alongside $X$?
   If the answer is "no", then the method could be generalized to any type of $Y$ (including discrete or mixed distributions),
   provided that $p(X)$ and $p(X \mid Y)$ still exist.
1. What is the difference between "NDoE, BNAF" and "BNAF"?
   Do I understand correctly that in "BNAF", $H(X)$ and $H(X \mid Y)$ are approximated via two separate flows?
1. How were the confidence intervals obtained?
1. NDoE, Real NVP is mentioned in Figure 5, but absent from the actual plot. Why?
1. There seem to be some minor issues with notation in Theorem B.1.
   Firstly, in the statement, $T$ is applied to $(y,x)$, but in the proof, the order is reversed:
   $U$ is applied to $(x,y)$.
   Of course, this is still perfectly valid; however, it introduces some unnecessary confusion.
   Secondly, from the notation $V = (T_1, \overline{T})$ it is not obvious that
   $\overline{T} \colon \mathbb{R}^{2n} \to \mathbb{R}^n$ (which, I assume, is implied here).

   Additionally, it seems that this theorem can be easily extended to the case of $X$ and $Y$ being of different dimensionalities.

   I kindly ask the authors to address my concerns regarding this theorem.
1. In Corollary B.2, $q$ is not defined.
   The role of $f = (f_1, f_2)$ is also not explained properly
   (as for the current state of the corollary, this can be any block-triangular normalizing flow).
   Please, clarify.
   I also suggest addressing the following notation conflict: $g$ in Corollary B.2 and on lines 742--743
   is not the same as on lines 751-755.
1. It might be due to the previously mentioned issues,
   but the following claim lacks rigorous backing:
   "Corollary B.2 suggests that given enough expressive power of our neural network architecture,
   we can (train?) the network to both approximate $H(X \mid Y)$ and $H(X)$".
   As we do not choose $g$ in Corollary B.2, it is not obvious that it is possible to achieve $\forall x,y \;\; g(y, f^x(x)) = f^x(x)$.
   Please, address my concerns and provide a more formal bridge between Corollary B.2 and the claim in question.

**Additional references:**

[1] Giulio Franzese et al. MINDE: Mutual information neural diffusion estimation. *Proc. of ICLR 2024.*

[2] Lee K., Rhee W. A Benchmark Suite for Evaluating Neural Mutual Information Estimators on Unstructured Datasets. *Proc. of NeurIPS 2024.*

---

> ### Author Response · Authors · 2024-11-27
>
> We appreciate the thoughtful and detailed feedback that helps improve the submission. Below are our comments on the review addressing the weaknesses pointed out.
>
> 1. We propose a framework of normalizing flows in estimating mutual information that can be adapted to any structure of normalizing flows and even more broadly generative models. We are mainly considering certain distributions that have typical properties (long tails, bounded, non-Gaussians). As the theory has suggested, the universal approximation of our method, challenges like manifold data can be considered as an extension of this and manifold aware normalizing flows can be considered to enhance the performance.
>
> 2. There were a few issues in the appendix, the presentation has been expanded and should hopefully make more sense.
>
> 3. The baselines are also combined with convolutional NN layers for feature extraction, which is a common practice. We understand that it is difficult to keep the comparison fair between discriminative and generative methods, thus we decide to choose what fits better to each others.
>
> 4. The method should in principle be applicable to other kinds of distributions, though model structure should be carefully selected. There are papers exploring discrete and mixed distributions for normalizing flows but we didn't go down that route in this paper.
>
> Regarding the minor concerns:
>
> Our work uses the DoE estimator as a theoretical base, the novelty is in implementing the two optimization problems simultaneously using normalizing flows. As for Ao\&Li, their approach is for the entropy and we argued that combining entropy estimators to obtain MI is not optimal, see the new plot in the paper. The other approaches are very recent (in particular, Butakov et al, 2024) and have appeared after our manuscript has been written. We plan to incorporate a detailed comparison in future work as part of downstream applications. Small issues with the plots have been taken care of. We are in the process of restructuring the source code to make it usable (this includes adding documentation).
>
> On the questions:
>
> Q1. This is a good point. While we haven't explored this in the example, we do believe that we should be able to generalize the method to any type of $Y$.
>
> Q2. Yes, two separate flows with same structure and parameters initialization.
>
> Q3.  We perform 10 runs and compute the std of the 10 estimates.
>
> Q4. NDoE, RealNVP fail to provide estimations for Sparse Gaussian case, which might due to the structure of RealNVP. However, we only implement a simple version as a flavor of image-friendly normalizing flows. It should work better with more sophisticated and specific models.
>
> Q5-Q7. Several issues in presentation and with the notation have been fixed in the appendix. In particular, $g$ has been renamed into $\overline{f}$ and the statement of the results has been adjusted. We added additional justification to the claim that we can (in principle) find a flow of the claimed form by a composition of two flows, for which we have universal approximation results.

---

> > ### Comment · Reviewer_USyt · 2024-12-03
> >
> > Dear authors,
> >
> > Thank you for your response.
> >
> > I understand that there are certain results on the universality property of normalizing flows, as well as specialized architectures to tackle the "manifold issue". However, in practice, there may be a trade-off between the manifold learning and the density learning. For examples, please, refer to the overview provided in [r1]. As your method relies on the accurate PDF estimation, the datasets proposed in my original review may pose a significant problem to your method. Therefore, I find it crucial to assess the proposed estimator on image-like data. The importance of such evaluation is also high due to distribution-free critic-based methods performing relatively well on such datasets.
> >
> > I am also moderately concerned by the fact that the general approach based on a single-model conditional & unconditional flow have not been considered instead of BNaF. I kindly suggest the authors to revise their work using a more general approach, as it will make the method applicable to a wider range of distributions and, probably, will simplify the theoretical part.
> >
> > Other than that, most of my concerns have been addressed. I will keep my score until I thoroughly read the revised theoretical part of the manuscript.
> >
> > [r1] Johann Brehmer, Kyle Cranmer. Flows for simultaneous manifold learning and density estimation

---

> > > ### Author Response · Authors · 2024-12-03
> > >
> > > We sincerely thank the reviewer for their insightful discussion. To conclude the conversation, and in case this may be of interest to the reviewer, we would like to point out that this issue is actually a general challenge in generative models. A helpful reference on this topic is the recent survey by Loaiza-Ganem (2024). While we acknowledge the limitation, a potential solution could involve estimating the MI for the learned representations instead, by combining our approach with other models, a strategy that has been shown to be feasible with our method.
> > >
> > > Furthermore, we agree that this is an excellent suggestion for generalizing our method. Although we were unable to implement this due to time constraints, we are eager to explore it in future work.

---

### Official Review · Reviewer_7bFp · 2024-11-05

**Soundness:** 2
**Presentation:** 1
**Contribution:** 1
**Rating:** 3
**Confidence:** 3

**Summary:**

The paper proposes a training regime for a block neural autoregressive flow (B-NAF) for difference-of-entropies (DoE) estimation of mutual information using a single normalizing flow (as an alternative to the naive implementation with two flows). The paper evaluates the method on several synthetic mutual information estimation tasks.

**Strengths:**

- Originality: to the best of my knowledge, simultaneous estimation of conditional and unconditional density using a single normalizing flow with application to mutual information estimation has not been explored in literature before.
- Quality: good narrative flow, generally consistent mathematical notation, clear figures. The paper is self-contained: the authors provide an extensive introduction to normalizing flows and mutual information estimation, including relevant prior work.

**Weaknesses:**

- Clarity: the proposed method is not presented with great clarity, and even after multiple reads I am not sure I understand _completely_ what the authors are proposing, and how it is motivated. In particular, it is not clear to me where the Eq. (11) comes from, and what the terms in it are precisely. The paragraph following Eq. (11) seems key to understand the idea, but is extremely dense and hard to parse. Authors repeatedly use the word "deactivate" (weights/sub-network), but don't explain precisely what it means. Algorithm 1 does not seem to optimize a part of the flow: $f_1$ (both losses only involve $f_2$). I suggest the authors shorten the Section 2 significantly (by e.g. moving parts to the appendix, or leaning more on prior work), and use the space to expand the Section 3, being more precise and clear when introducing and motivating their method.
- Significance: the benefit of using a single flow (as proposed, i.e. NDoE, BNAF) instead of two flows (BNAF) is not clear from the results presented. While authors claim that "proposed model achieved better performance across different dimensionalities and sample sizes", looking at Figures 2-5 I see, at most, a marginal improvement of NDoE, BNAF over BNAF, and often no improvement at all. The significance would be clearer if authors quantified the (relative/absolute) improvement in text, and provided an argument as to why it's significant (avoiding phrases like "_slight_ bias"). Moreover, authors only report results on synthetic data in the main text: if experiments were run on real data, authors should at least summarize the findings in the main text. Finally, in the conclusion authors say that they "plan to evaluate our method in view of downstream applications that require computation of mutual information" -- expanding the introduction to include a paragraph on what the most important applications of mutual information estimation are would further showcase significance.

**Questions:**

- Where does Eq. (11) come from precisely? Why does it not include the derivative of $f_2$ w.r.t $y$?
- How do we train $f_1$ using Algorithm 1?
- What do authors mean when they say "deactivate the off-diagonal weights" in Algorithm 1?
- What can authors say about the computational cost of the method compared to e.g. BNAF, i.e. training separate flows?

---

> ### Author Response · Authors · 2024-11-27
>
> The concerns of the reviewers are related to clarity and significance.
>
> * Clarity: we restructured the derivation of the method and added more detail surrounding Eq (11). It is true that $f_1$ does not play a role here (except conceptually) and we optimize a particular form of $f_2$ that would, by setting certain weights to zero, 'switch' between a parametrization for H(X) and one for H(X|Y). The deactivation of off-diagonal weights means simply masking them out during training (or setting them to zero).
>
> * Significance:  Theoretically speaking, we moved two steps forward from DoE estimator(McAllester\&Stratos, 2018). Firstly, the DoE estimator is still essentially a set of entropy estimators of $H(x)$ and $H(x|y)$. Our method is a mutual information estimator at the same time it can serve as entropy estimators. On the other hand, one can freeze the parameters of the trained $p(x)$ model and add on conditional NNs / Adapters for any $Y$. This allows its implementation for fine-tuning and specifying pre-trained models.
>
> Experimentally speaking, although estimating via separate flows (BNAF) and our method will eventually converge to true MI with enough expressive power and careful training according to universal approximation theory, our method has early convergence especially when samplesize is small. We included additional plots illustrating the faster convergence (Appendix C.4). Another feature of our method is that its estimation has constraints of $I(X,Y)>=0$ and it has the similar behavior of a upper bound estimator. This provide a good estimation of small true MI, which is considered a limitation of generative models in this area before. We believe that this may be related to the correlation boosting effect proposed by (Gao et. al, 2018).

---

### Official Review · Reviewer_cMNr · 2024-11-06

**Soundness:** 1
**Presentation:** 2
**Contribution:** 1
**Rating:** 3
**Confidence:** 3

**Summary:**

This paper explores block neural autoregressive normalizing flows (B-NAF) for estimating the mutual information (MI) between two continuous random variables, denoted X and Y, from samples. The idea is to decompose MI as the difference between H(X) and H(X|Y), respectively denoting the entropy of X and the conditional entropy of X given Y. The observes that the (conditional) entropy is a direct by-product of density estimation with the KL divergence as $H(X) = \inf_{q} \mathbb -{E}_{p(x)}[\log q(x)] $. By exploiting the autoregressive nature of B-NAF we can jointly estimate H(X) and H(X|Y) with one B-NAF network.

**Strengths:**

- Using normalizing flows, in particular autoregressive flows, to do MI estimation is a novel idea to the best of my knowledge.
- Empirical results demonstrate that the proposed method can achieve good results on simple benchmarks.

**Weaknesses:**

- Presentation: Overall I do not find the paper very well-written. The problem is not well-motivated and it remains unclear under which setting the proposed approach can be of practical usefulness. Section 2 spans from page 2 to page 6 and does not contain original results in my opinion. In contrast, section 3 is very short and is more difficult to read whereas it is probably the most important part of the paper. Figures are very hard to read with almost 19 coloured lines on each plot -- it is unclear to me what is the value of such plot for the reader and message of the paper. Finally there is no conclusion or discussion of future work, potential impact, weaknesses or whatsoever of the paper.
- Soundness: Overall I have trouble to really understanding the practical relevance of estimating MI from samples as this ends up being equivalent to density estimation. It is thus very sensitive to the choice of model class and, in my opinion, inspecting MI depends as much on the model class chosen than on the samples. This is particularly true for higher dimensional problem where density estimation becomes intractable without strong modelling assumptions. I agree my statement is strong and there may exist certain usecases where estimating MI with minimal modelling assumptions can be relevant, however the paper fails to motivate such use cases and to demonstrate the value of the proposed approach for such settings.
- Novelty: Normalizing flows perform density estimation while providing both density evaluation and sample generation. It is also clear and well known that the MI can be estimated by sampling and evaluating p(x, y) (potentially by exploiting Bayes' rule to decompose p(x,y) into factors). Thus I do not find the idea presented in this paper very novel and I can imagine many researcher have already used NF to estimate MI when they felt it was a useful value to look at.

**Questions:**

I do not understand why you couldn't simply learn a B-NAF (implicitly decomposing p(x, y) as p(x), p(y|x)) with standard MLE of the joint distribution of x, y. What is the gain of alternating between two optimization problem whereas directly solving density estimation with the right architecture would do the same job. Can you provide some motivation and why don't you compare to that more natural approach?

---

> ### Author Response · Authors · 2024-11-27
>
> We would like to thank the reviewer for the comments. Regarding the presentation, we improved the plots and hope that they are more readable now. We also expanded section 3 to include more details and expanded the conclusions section slightly to discuss limitations and further work.
>
> The reviewer's main concerns seem to be in relation to the relevance of the problem and our approach to it. A growing body of work on mutual information estimation is motivated by, among other things, the limitations of classical density estimators in high dimension. We acknowledge that we may not have include enough justification for this, and included further references, as these issues are already discussed at length in the existing literature.
>
> Finally, the question appears to be based on a misunderstanding. Of course, it would be easy to learn $p(x)$ and $p(y|x)$ simultaneously, but here we want to learn $p(x)$ and $p(x|y)$. Learning the joint entropy and the individual marginal entropies and combining them to a MI estimator is not only less efficient (we illustrate this in a new graph), but also introduces bias, as mentioned in Section 1.1.

---

> > ### Comment · Reviewer_cMNr · 2024-11-28
> >
> > Thank you for your response and updating a new version of the manuscript.
> >
> > > Regarding the presentation, we improved the plots and hope that they are more readable now. We also expanded section 3 to include more details and expanded the conclusions section slightly to discuss limitations and further work.
> >
> > I appreciate the efforts you’ve made to improve the readability of the plots and to expand Section 3 and the conclusions. The figures are indeed clearer, though I still find them somewhat crowded. Presenting four figures, each containing six plots, appears overwhelming and potentially diminishes the key points being conveyed. It might be possible to consolidate these into fewer, more focused figures while retaining the thoroughness of your analysis. I would also appreciate more diverse, and real-world, benchmarks than the one considered.
> >
> > >The reviewer's main concerns seem to be in relation to the relevance of the problem and our approach to it. A growing body of work on mutual information estimation is motivated by, among other things, the limitations of classical density estimators in high dimension. We acknowledge that we may not have include enough justification for this, and included further references, as these issues are already discussed at length in the existing literature.
> >
> > Your response references the "limitations of classical density estimators in high dimensions" as motivation for mutual information estimation. However, the connection between these two concepts is unclear to me. Did you add a discussion elaborating on this in the updated manuscript? If so, could you highlight the relevant modifications? As a suggestion, using a different color or a marked-up version of the paper to highlight the changes could facilitate the review process and make updates more transparent.
> >
> > > Finally, the question appears to be based on a misunderstanding. Of course, it would be easy to learn $p(x)$ and $p(y|x)$ simultaneously, but here we want to learn $p(x)$ and $p(x|y)$. Learning the joint entropy and the individual marginal entropies and combining them to a MI estimator is not only less efficient (we illustrate this in a new graph), but also introduces bias, as mentioned in Section 1.1.
> >
> > Thank you for clarifying my earlier misunderstanding regarding the differences between estimating p(x) and p(x∣y) versus p(x) and p(y∣x). Could you elaborate on why combining joint and marginal entropies introduces bias, while your proposed method avoids this issue?
> >
> >
> > I remain somewhat skeptical about the broader relevance and originality of this work, as it appears to leverage well-known properties of normalizing flows (e.g., KL/log-likelihood-based training). Additionally, while the methodology is interesting, I still find the presentation rough and feel there is room for improvement in clearly articulating the contributions.
> >
> > As a practical remark, it is well-established that normalizing flows become unstable when data resides on a lower-dimensional manifold than the flow’s dimensionality. Since mutual information itself is ill-defined when the density p(x,y) is poorly behaved, how does this impact the practical relevance of your work? Addressing this limitation or including a discussion on potential mitigation strategies would also something I would find interesting.
> >
> > Thank you again for engaging with the review process. I am happy to discuss these points further and look forward to your response.

---

> > > ### Author Response · Authors · 2024-11-28
> > >
> > > Thank you for your prompt and insightful feedback. While time constraints may limit our ability to incorporate all suggestions on the presentation in this revision, we assure you that we will include them before making the paper public.
> > >
> > > In our related work section, we indeed discuss the limitations of these estimators as motivation for the recent developments in neural estimators, including both discriminative (e.g., MINE) and generative approaches. Regarding normalizing flows in particular, estimating $H(X)$, $H(Y)$, and $H(X,Y)$ separately introduces individual biases ($b_X$, $b_Y$, $b_{X,Y}$) that compound. This point aligns with observations made by Paninski (2003), which we reference in our related work section. In contrast, our approach of jointly estimating $H(X)$ and $H(X|Y)$ leads to bias terms that cancel out due to shared parameters.
> > > We have added a figure in Appendix C.5 to illustrate the convergence properties of our method. An additional feature of our approach is that our mutual information estimates remain consistently positive during training, which is not guaranteed by other methods.
> > >
> > > We acknowledge your point about normalizing flows on manifolds, ideally they should map to manifolds of equal dimension. While an in-depth exploration of regularization issues in such settings is beyond the scope of this work, we agree that it presents an interesting avenue for future research. We would be happy to include brief comments on this topic in our paper to highlight its potential significance.

---

> > > > ### Comment · Reviewer_cMNr · 2024-12-03
> > > >
> > > > I thank you for your further response.
> > > >
> > > > Unfortunately, I still find the line of arguments regarding bias very hand-wavy and would only be convinced by it with strong theoretical or empirical evidence.
> > > >
> > > > I do not really believe my major concerns (spanning the three main evaluation criteria: presentation, soundness & contribution) have been addressed. In particular,
> > > > - I have not seen a meaningful motivation for the setup considered;
> > > > - The empirical evaluation remains very synthetic and does not really enlighten the reader regarding the properties of the proposed estimator.
> > > >
> > > > For this reason, I will keep my score.

---

### Comment · Area_Chair_n12r · 2024-11-26

Dear all,

The deadline for the authors-reviewers phase is approaching (December 2).

@For reviewers, please read, acknowledge and possibly further discuss the authors' responses to your comments. While decisions do not need to be made at this stage, please make sure to reevaluate your score in light of the authors' responses and of the discussion.

- You can increase your score if you feel that the authors have addressed your concerns and the paper is now stronger.
- You can decrease your score if you have new concerns that have not been addressed by the authors.
- You can keep your score if you feel that the authors have not addressed your concerns or that remaining concerns are critical.

Importantly, you are not expected to update your score. Nevertheless, to reach fair and informed decisions, you should make sure that your score reflects the quality of the paper as you see it now. Your review (either positive or negative) should be based on factual arguments rather than opinions. In particular, if the authors have successfully answered most of your initial concerns, your score should reflect this, as it otherwise means that your initial score was not entirely grounded by the arguments you provided in your review. Ponder whether the paper makes valuable scientific contributions from which the ICLR community could benefit, over subjective preferences or unreasonable expectations.

@For authors, please respond to remaining concerns and questions raised by the reviewers. Make sure to provide short and clear answers. If needed, you can also update the PDF of the paper to reflect changes in the text. Please note however that reviewers are not expected to re-review the paper, so your response should ideally be self-contained.

The AC.

---

### Meta-Review · Area_Chair_n12r · 2024-12-20

**Metareview:**

The reviewers recommend rejection (3-3-5-5-8-5). The paper proposes a new mutual information estimator based on conditional normalizing flows. The approach is well-motivated, but the results are not convincing when compared to BNAF. The author-reviewer discussion has been constructive and has led to a number of clarifications and improvements. However, the reviewers still question the motivation, soundness, novelty, and significance of the results. For these reasons, I recommend rejection. I encourage the authors to address the reviewers' comments and to resubmit to a future conference.

**Additional Comments On Reviewer Discussion:**

The reviewers are quite unanimous in their recommendations. The discussion was not sufficient to significantly change the opinions of the reviewers.

---

### Decision · Program_Chairs · 2025-01-22

Reject